# Human Nup98 regulates the localization and activity of DExH/D-box helicase DHX9

Juliana S Capitanio[1], Ben Montpetit[1,2], Richard W Wozniak[1]*

[1]Department of Cell Biology, University of Alberta, Edmonton, Canada; [2]Department of Viticulture and Enology, University of California, Davis, United states

**Abstract** Beyond their role at nuclear pore complexes, some nucleoporins function in the nucleoplasm. One such nucleoporin, Nup98, binds chromatin and regulates gene expression. To gain insight into how Nup98 contributes to this process, we focused on identifying novel binding partners and understanding the significance of these interactions. Here we report on the identification of the DExH/D-box helicase DHX9 as an intranuclear Nup98 binding partner. Various results, including in vitro assays, show that the FG/GLFG region of Nup98 binds to N- and C-terminal regions of DHX9 in an RNA facilitated manner. Importantly, binding of Nup98 stimulates the ATPase activity of DHX9, and a transcriptional reporter assay suggests Nup98 supports DHX9-stimulated transcription. Consistent with these observations, our analysis revealed that Nup98 and DHX9 bind interdependently to similar gene loci and their transcripts. Based on our results, we propose that Nup98 functions as a co-factor that regulates DHX9 and, potentially, other RNA helicases.

*For correspondence: rick. wozniak@ualberta.ca

**Competing interests:** The authors declare that no competing interests exist.

## Introduction

The nuclear envelope (NE) forms a physical barrier between the DNA-containing nucleoplasm and the cytoplasm. This membrane system in conjunction with macromolecular gateways, termed nuclear pore complexes (NPCs), regulate transport across the NE. Highly conserved across all eukaryotes, NPCs are composed of ~30 proteins, termed nucleoporins or Nups, which can be placed into two general categories. One set of Nups lie at or near the membrane and form the cylindrical eight-fold symmetrical NPC scaffold. Among its functions, the core scaffold acts as a binding surface for a second set of Nups (FG-Nups) that line the NPC channel and directly facilitate the movement of nuclear transport factors and their cargoes through the NPC (*Field et al., 2014*).

While the roles of Nups in NPC structure and nuclear transport have been well established, numerous observations indicate that Nups also function outside of NPCs in the cytoplasm and nucleoplasm (*Chatel and Fahrenkrog, 2012*; *Hou and Corces, 2010*; *Ptak and Wozniak, 2016*; *Raices and D'Angelo, 2012*). For example, various FG-Nups have been detected in the nucleoplasm, which have been shown to move between intranuclear sites and NPCs (*Rabut et al., 2004*). In addition to contributing to nuclear transport (*Sakiyama et al., 2016*; *Zahn et al., 2016*), these intranuclear Nups have been reported to regulate gene expression through binding transcription sites (*Capelson et al., 2010*; *Kalverda et al., 2010*; *Ptak et al., 2014*), including immune response genes (*Faria et al., 2006*; *Light et al., 2013*; *Panda et al., 2014*), and influencing chromatin organization (*Kalverda and Fornerod, 2010*; *Liang and Hetzer, 2011*; *Ptak and Wozniak, 2016*).

Among the most studied Nups exhibiting intranuclear localization is Nup98 (*Griffis et al., 2002*; *Iwamoto et al., 2010*; *Radu et al., 1995*). Nup98 binds to the mRNA export factors Rae1 and NXF1

(TAP) and it has been shown to mediate mRNA export (*Bachi et al., 2000*; *Blevins et al., 2003*; *Powers et al., 1997*). Nup98 also participates in nuclear import and export of proteins though its interaction with importin-β family members (*Allen et al., 2001*) and the exportin CRM1 (*Oka et al., 2010*). Several distinct domains are present in Nup98, including an N-terminal region containing FG/GLFG repeats, a putative RNA-interacting domain, a binding site for Rae1, and a C-terminal region that interacts with other Nups (*Chatel and Fahrenkrog, 2012*; *Sun and Guo, 2008*).

Immunofluorescence analysis revealed that Nup98 is visible throughout the nucleoplasm, but can accumulate at intranuclear structures termed GLFG bodies (*Griffis et al., 2002*). Further, the mobility of NPC-associated and nucleoplasmic Nup98 is dependent on ongoing transcription in the cell (*Griffis et al., 2002*). Studies in Drosophila revealed the association of Nup98 with actively transcribed genes, especially those involved in development and cell cycle regulation, with modulation of cellular Nup98 levels (over-expression or knock-down) altering transcription of these genes (*Capelson et al., 2010*; *Kalverda et al., 2010*). Similarly, in mammalian cells, Nup98 has been shown to associate with chromatin and regulate gene expression during the differentiation of embryonic stem cells into neural progenitor cells (*Liang et al., 2013*). In these cells, Nup98 preferentially associates with the promoter regions of developmentally regulated genes, and changes in the levels of Nup98 are again found to alter gene expression.

Several recent observations have provided further insight into the role of Nup98 in transcription. Light and colleagues showed that mammalian Nup98 binds to *HLA-DRA* (*Light et al., 2013*), an interferon-γ induced gene exhibiting transcriptional memory (i.e. a gene that displays rapid induction given a recent history of being activated). In the absence of Nup98, transcriptional memory was lost and the binding of RNA polymerase II at promoters poised for reactivation was reduced, which matched similar findings in yeast (*Light et al., 2010*, *2013*). In Drosophila, Pascual-Garcia and colleagues showed binding of Nup98 to the promoter regions of certain active genes and a requirement for Nup98 in their transcription (*Pascual-Garcia et al., 2014*). Nup98 binding to these genes was dependent on TRX and MBD-R2, a component of the NSL (nonspecific lethal) complex that directs histone H4K16 acetylation. However, the loss of Nup98 did not change H4K16 acetylation or TRX-mediated H3K4 trimethylation patterns, both of which are required for active transcription and transcriptional memory. Thus, the function of Nup98 in the transcription of these loci remains unclear.

More evidence for the role of Nup98 in gene expression regulation comes from studies of hematopoietic malignancies. More than twenty-eight different chromosomal rearrangements involving the *NUP98* gene have been identified. The resulting fusion proteins have been shown to alter transcription through fusing the N-terminal domain of Nup98 (*Bai et al., 2006*; *Kasper et al., 1999*) to a C-terminal domain that usually contains a chromatin/DNA interacting region (*Capitanio and Wozniak, 2012*). The oncogenicity of several Nup98 fusions has been demonstrated in mouse models where Nup98 fusions lead to acute myeloid leukemia recapitulating the human disease phenotype (*Gough et al., 2011*; *Moore et al., 2007*). Finally, Nup98 also impacts gene expression at the post-transcriptional level. A recent publication reported that Nup98 associates with the p21 mRNA 3'UTR preventing degradation by the exosome, with several other putative target mRNAs being similarly regulated (*Singer et al., 2012*).

Despite the growing evidence linking Nup98 to the regulation of chromatin structure and gene expression, little is known about the mechanism by which Nup98 affects these processes. In this study, we have focused on identifying novel Nup98 binding partners and assembling a Nup98 interaction network. Of the Nup98 interactors, one of the strongest binding partners was the DExH/D-box protein DHX9 (RNA helicase A). We demonstrate that Nup98 binds DHX9 in the nucleoplasm, regulates the nuclear distribution of DHX9, and influences DHX9 RNA-binding and ATPase activity. Their interactions ultimately influence gene expression at the level of DHX9-mediated transcription and splicing. These data provide evidence for a novel mechanism by which the nucleoporin Nup98 can regulate gene expression away from NPCs.

## Results

### Identification of Nup98 interacting partners

Nup98 is a component of NPCs, but it has also been shown to reside in the cytoplasm and nucleoplasm (*Griffis et al., 2002*). The presence of this Nup in different locations likely reflects the participation of Nup98 in distinct cellular processes. To further understand these putative non-NPC functions, we focused on identifying Nup98 binding partners. *GFP-NUP98* or *GFP* alone was expressed in HEK293T cells and immunoprecipitated (IP) using antibodies directed against GFP. Mass spectrometry (MS) analysis of purified protein complexes (*Figure 1*, *Supplementary file 1A*) identified previously characterized Nup98 interactors, such as Nup88 (*Griffis et al., 2003*), Rae1 (*Pritchard et al., 1999*), NXF1 (*Bachi et al., 2000*), and CRM1 (*Oka et al., 2010*), as well as several other proteins.

Nup98-interacting proteins were prioritized for analysis based on the number of unique peptides mapped to the protein, the percent coverage of the protein sequence (*Liu et al., 2004*), and absence of the protein in a database of common contaminants identified by IP-MS (*Mellacheruvu et al., 2013*). Nup98 interactors were further annotated with curated protein-protein interactions (PPI) to create a PPI network (*Figure 2A*). Network clustering identified highly interconnected nodes within the network, possibly reflecting protein complexes that may interact with GFP-Nup98 within the context of distinct cellular processes. Enrichment analysis based on Gene Ontology (GO) annotations indeed showed that these sets of GFP-Nup98 interacting proteins function in specific mRNA metabolism events including mRNA processing, splicing, stabilization, and transport (*Figure 2A*).

We speculated that proteins of the highest abundance (i.e. highest number of unique peptides detected by LC-MS/MS) are likely the nearest neighbors of Nup98 (*Mazloom et al., 2011*), and proteins with the highest number of PPIs in the network (i.e. hubs) may represent key components that interact with Nup98 within the context of these processes (*He and Zhang, 2006*). Therefore, we selected Nup98 interactors for further study on the basis of abundance in the GFP-Nup98 immunoprecipitation (IP) and the number of PPIs occurring with the other proteins within the network (*Figure 2A and B*). Based on these features, DHX9 and hnRNP U were selected for further analysis. Importantly, the association of both DHX9 and hnRNP U with Nup98 was confirmed by reciprocal immunoprecipitations of endogenous Nup98, DHX9, and hnRNP U (*Figure 2—figure supplement 1*).

### Nup98 influences the intranuclear distribution of DHX9

Both DHX9 and hnRNP U are RNA-binding proteins that reside in the nucleoplasm (*Dreyfuss et al., 1984*; *Uhlén et al., 2015*; *Weidensdorfer et al., 2009*; *Zhang and Grosse, 1991*). We compared the localization of these proteins and Nup98 in HEK293T cells using immunofluorescence microscopy (*Figure 3*). Similar to previous reports (*Uhlen et al., 2010*; *Uhlén et al., 2015*), DHX9 and hnRNP U were broadly distributed within the nucleoplasm in a punctate pattern, but appeared excluded from nucleoli. Neither protein appeared concentrated at the NE (*Figure 3*, inset). Nup98 was also detected within the nucleoplasm (*Figure 3*); however, the broad intranuclear distributions of Nup98, DHX9, and hnRNP U made it difficult to judge the significance of any signal overlap. As an alternative approach to assess the physical relationship between these proteins, we examined the consequences of depleting or overproducing Nup98 on the nuclear distribution of DHX9 and hnRNP U. Depletion of Nup98 resulted in no detectable changes in the distribution pattern of hnRNP U (*Figure 4*). However, the loss of Nup98 caused the appearance of bright DHX9 foci in the nucleoplasm. Moreover, the exclusion of DHX9 from nucleoli observed in mock-treated cells was less pronounced in Nup98-depleted cells, suggesting that DHX9 has greater access to the nucleolus in the absence of Nup98. In contrast, depletion of DHX9 (or hnRNP U; data not shown) did not noticeably alter Nup98 localization (*Figure 4—figure supplement 1*).

Combined with the protein-interaction data, these results are consistent with a model in which Nup98 contributes to the steady-state localization of DHX9 within the nucleoplasm. To further test this idea, we increased cellular levels of Nup98 and examined the distribution of DHX9 (*Figure 5A*). Elevated levels of Nup98 accumulate in intranuclear foci termed GLFG-bodies (*Griffis et al., 2002*), and we observed the formation of these foci in cells producing GFP-Nup98 (*Figure 5A*). Importantly,

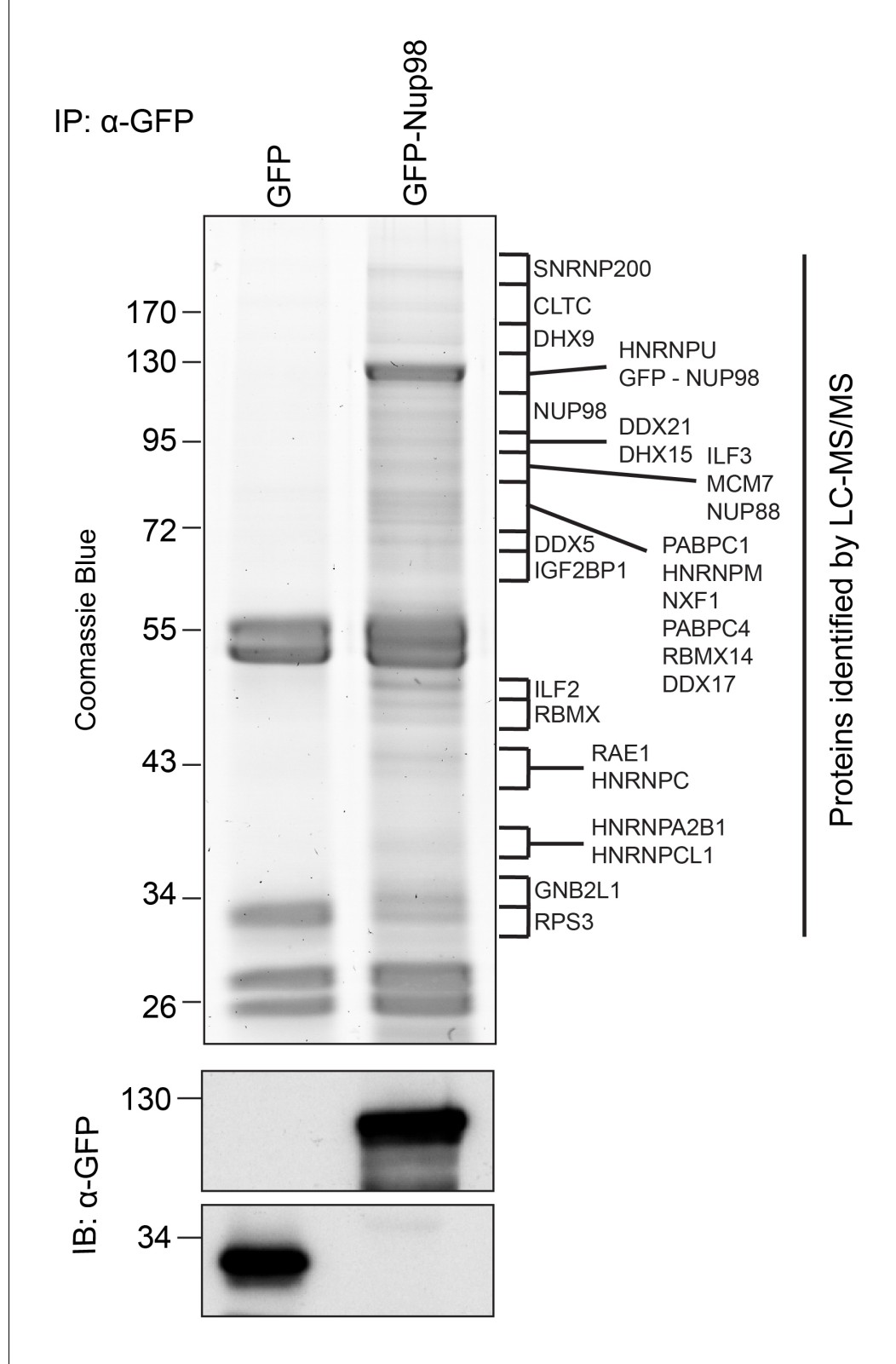

**Figure 1.** Identification of Nup98-interacting proteins. Plasmids encoding GFP-Nup98 or GFP alone were transfected into HEK293T cells. These proteins were immunoprecipitated from whole cell lysates using an antibody directed against GFP. Co-immunoprecipitated proteins were analyzed by SDS-PAGE and gel pieces containing regions of interest were analyzed by LC-MS/MS to identify proteins co-immunoprecipitated with GFP-Nup98. Western blotting of these fractions using anti-GFP antibodies confirmed the presence of GFP and GFP-Nup98 (bottom panel). Protein species indicated

*Figure 1 continued on next page*

*Figure 1 continued*

on the right of the gel represent those producing peptides most frequently identified by LC-MS/MS in the GFP-Nup98 immunoprecipitated fractions. The positions of molecular mass markers (shown in kDa) are indicated on the left.

DHX9 was recruited to the GFP-Nup98 foci. By contrast, no visible impact on hnRNP U distribution was observed in GFP-Nup98 producing cells (*Figure 5—figure supplement 1*). The change in DHX9 distribution was not accompanied by alterations in the cellular levels of DHX9 (*Figure 5—figure supplement 2*), thus the DHX9 associated with GFP-Nup98 foci was likely recruited from other locations.

Our observation that DHX9 is localized to intranuclear Nup98-containing foci and did not appear to accumulate at NPCs with Nup98 (*Figure 3*) suggests that these proteins interact in the nucleoplasm. To further test this model, HeLa cell nuclei were fractionated to make nucleoplasmic and NE enriched fractions, which could be used to further assess the location of the Nup98-DHX9 interaction. Consistent with localization data, DHX9 was primarily present in a nucleoplasmic fraction, while Nup98 was detected in both the nucleoplasmic and NE fractions, which were further validated with antibodies against other NE and nucleoplasmic proteins (*Figure 5B*). Immunoprecipitations from these fractions showed that DHX9 was only detected in association with nucleoplasmic Nup98, although similar amounts of Nup98 were purified from both fractions, and that nucleoplasmic DHX9 was able to immunopurify Nup98 (*Figure 5C*).

We also examined the interactions of DHX9 with an N-terminal region of Nup98 (GFP-Nup98$^{1-497}$). When expressed in cells, GFP-Nup98$^{1-497}$ can enter the nucleoplasm and induce the formation of GLFG bodies, however it does not associate with NPCs (*Griffis et al., 2002*; *Kalverda et al., 2010*). As observed with the full-length GFP-Nup98, DHX9 was recruited to GLFG-bodies formed by GFP-Nup98$^{1-497}$ truncation (*Figure 5—figure supplement 3A*). Consistent with this result, DHX9 was detected in association with immunopurification of GFP-Nup98$^{1-497}$, but not a C-terminal fragment, GFP-Nup98$^{498-920}$ (*Figure 5—figure supplement 3B*). Cumulatively, these results strongly argue that the Nup98-DHX9 complex is primarily present in the nucleoplasm, and that the N-terminal FG/GLFG domain of Nup98 interacts with DHX9.

## Binding of Nup98 to DHX9 is enhanced by RNA

Since DHX9 and Nup98 both interact with RNA (*Fuller-Pace, 2006*; *Ren et al., 2010*), we also investigated the importance of RNA in their association. In Nup98 immunoprecipitates from HEK293T cell lysates we detected DHX9, PRKDC, and several Nups bound to Nup98 (*Figure 6A*), consistent with the results presented in *Figure 1*. However, when parallel samples of bead-bound complexes were incubated with RNase A in amounts sufficient to degrade all detectable RNA (*Höck et al., 2007*; *Moore et al., 2014*; *Ule et al., 2005*; *Zhang et al., 2008*), levels of Nup98-associated DHX9 were reduced, while PRKDC and associated Nups were unaffected. These results imply that RNA, directly or indirectly, contributes to the interaction of Nup98 with DHX9.

To extend our characterization of the in vivo interactions between Nup98, DHX9, and RNA, we examined whether recombinant Nup98 and DHX9 could interact using in vitro binding assays. Magnetic beads coupled to anti-DHX9 antibodies were incubated with GST-DHX9. GST alone or GST-Nup98 was then added to bead-bound GST-DHX9. Only the GST-Nup98 protein bound to the beads, suggesting a direct interaction between DHX9 and Nup98. (*Figure 6B*). Similar results were obtained using DHX9 and Nup98 lacking the GST tags (*Figure 6—figure supplement 1A*). As it was possible that RNA present within the *E. coli* extracts could contribute to the in vitro DHX9-Nup98 interaction, we also conducted binding reactions after pre-treating the recombinant proteins with RNase A (to remove any residual RNA) or adding excess RNA (poly I:C). Binding reactions conducted under these conditions reveal that the interaction of untagged or GST-tagged DHX9 with Nup98 was partially reduced by the addition of RNase A, while the addition of RNA (poly I:C) did not appear to alter the binding of DHX9 to Nup98 (*Figure 6B* and *Figure 6—figure supplement 1A*).

A second in vitro assay previously employed to assess Nup-Nup interactions (*Patel and Rexach, 2008*; *Zhou et al., 2013*) was also used to evaluate the interaction of Nup98 with DHX9. Termed the 'bead halo' assay, protein (e.g. DHX9) is bound to beads and then incubated with a potential binding partner (Nup98). Binding of Nup98 to bead-associated DHX9 is detected using anti-Nup98

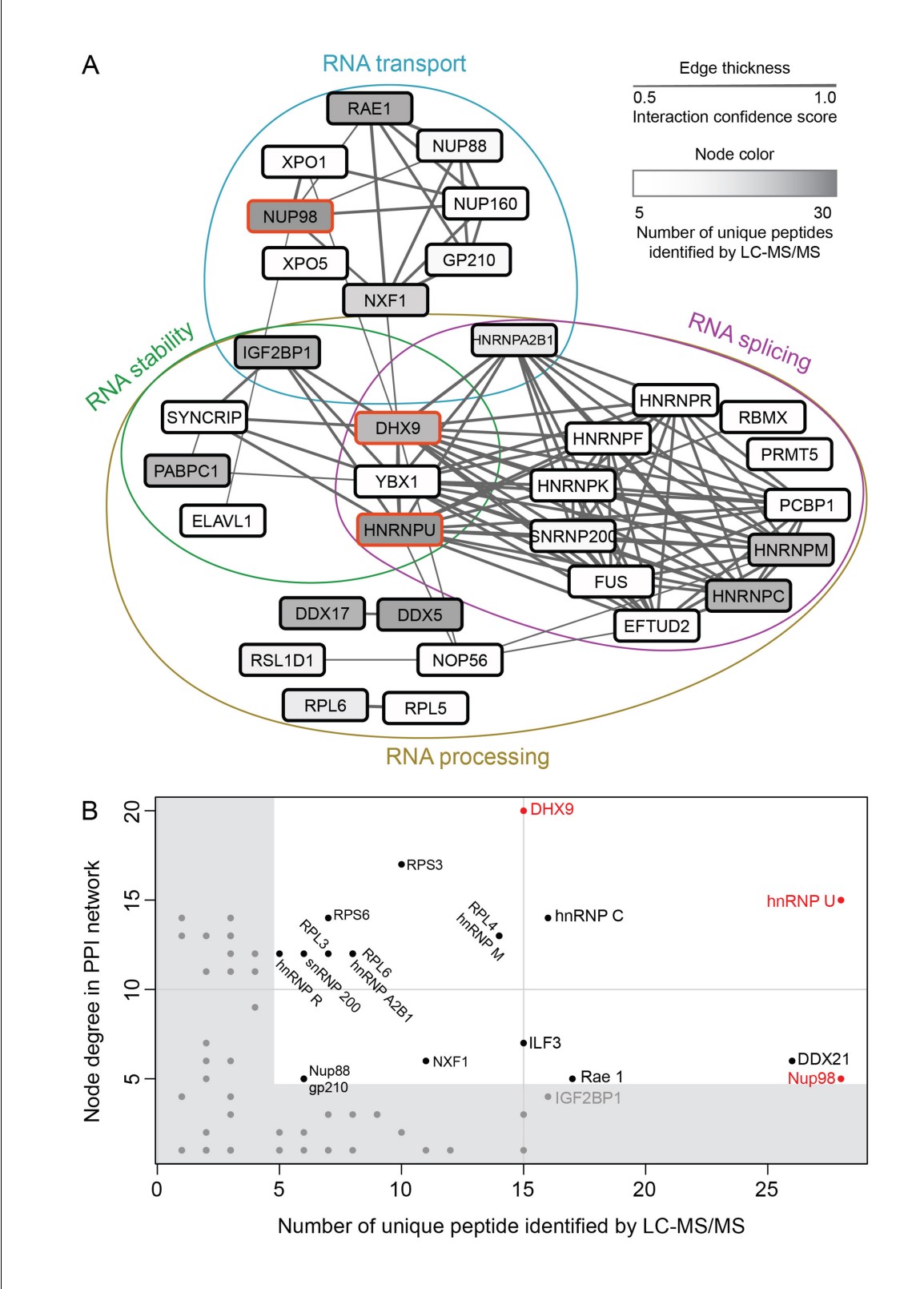

**Figure 2.** Nup98 protein-protein interaction network identified by IP-MS. (**A**) Curated protein-protein interactions (PPI) among identified Nup98 binding partners are represented in a PPI network. Edge thickness indicates the confidence score for the interaction and node color indicates abundance of the interactor in the GFP-Nup98 immunoprecipitation. Biological functions of identified protein complexes are indicated in the colored Venn diagram superimposed on the network. Nup98, DHX9 and hnRNP U are indicated by a red border. (**B**) Nup98 interactors were prioritized based on their degree
*Figure 2 continued on next page*

*Figure 2 continued*

of interconnection and the number of unique peptides identified by MS. In the scatterplot, node degree in the PPI network (y-axis) identifies hubs in the GFP-Nup98 PPI network, while the number of unique peptides (x-axis) reflects the abundance of the indicated protein in the purified GFP-Nup98 protein complex. Nup98 and its interactors DHX9 and hnRNP U are shown in red.

The following figure supplement is available for figure 2:

**Figure supplement 1.** Immunoprecipitation of endogenous Nup98 with DHX9 and hnRNP U.

antibodies and fluorescently labeled secondary antibodies. Interactions between the proteins are visualized by a fluorescent signal on the surface of the beads (*Figure 6—figure supplement 1B*). The level of bead-associated fluorescence signal provides a relative measure of the strength of the interaction (*Patel and Rexach, 2008*; *Zhou et al., 2013*). Using this assay, we detected and quantified the binding of recombinant Nup98 to DHX9 (*Figure 6C* and *Figure 6—figure supplement 1B*), and again the addition of RNA did not significantly alter the relative strength of this interaction, but the inclusion of RNase A reduced the level of DHX9 binding to Nup98 (*Figure 6C* and *Figure 6—figure supplement 1B*).

The bead halo assay was also used to identify regions of DHX9 that interact with Nup98. GST-Nup98 bound to beads was incubated with three consecutive, non overlapping domains of DHX9 tagged with GFP (*Figure 6D*). We observed that an N-terminal region (residues 1–380), containing two double-stranded RNA binding motifs (DRBM1 and DRBM2), and a C-terminal segment (residues 821–1270), containing an (OB)-binding fold and a single-stranded RNA-binding RGG-box, bound to Nup98. Conversely, DHX9's central region (residues 381–820), containing its ATP-dependent helicase domain (*Zhang and Grosse, 1997*), did not bind GST-Nup98 under these conditions (*Figure 6D* and *Figure 6—figure supplement 1C*). The interactions of the N- and C-terminal domains of DHX9-GFP with GST-Nup98 appeared to be facilitated by the presence of RNA, as these interactions were sensitive to RNase A. Furthermore, the addition of RNA (poly I:C) prior to mixing of the two proteins stimulated binding of GST-Nup98 and the N-terminal domain of DHX9-GFP (*Figure 6D*). Cumulatively, these data lead us to conclude that DHX9 can directly bind Nup98 and that their association is augmented by RNA. The DHX9-Nup98 interaction is likely mediated by the N- and C-terminal domains of DHX9.

## Nup98 stimulates the ATPase activity of DHX9

Like other RNA helicases, DHX9 can bind and hydrolyze ATP, which can promote unwinding of duplex RNA and remodelling of RNA–protein complexes (*Fullam and Schröder, 2013*; *Fuller-Pace, 2006*; *Zhang and Grosse, 1994*). Binding partners of RNA helicases have been shown to regulate helicase function by inhibiting or stimulating their ATPase activity (*Bourgeois et al., 2016*) and we hypothesized that Nup98 could play a similar role with DHX9. To test this, the ATPase activity of recombinant GST-DHX9 was examined in the presence and absence of GST-Nup98. We observed a basal ATPase rate for recombinant DHX9 that was stimulated by the addition of RNA to levels comparable to that previously reported for DHX9 (*Figure 7—figure supplement 1A*) (*Schütz et al., 2010*; *Zhang and Grosse, 1994*; *1997*). In the presence of excess RNA (poly I:C), the addition of GST-Nup98 induced a dose-dependent increase in GST-DHX9 ATPase activity reaching levels approximately five-fold higher than GST-DHX9 and RNA alone at the highest GST-Nup98 concentration tested (*Figure 7*). A similar level of stimulation was also observed using untagged versions of DHX9 and Nup98 (*Figure 7—figure supplement 1B*). DHX9 ATPase activity was also increased upon addition of the N-terminal FG/GLFG domain of Nup98. By contrast, neither the C-terminal domain of Nup98 or GST alone caused significant changes in DHX9 ATPase activity, nor could GST-Nup98 stimulate GST-DHX9 in the absence of RNA (*Figure 7—figure supplement 1C*). These data indicate that Nup98 functions as a positive regulator of DHX9 ATPase activity in the context of RNA.

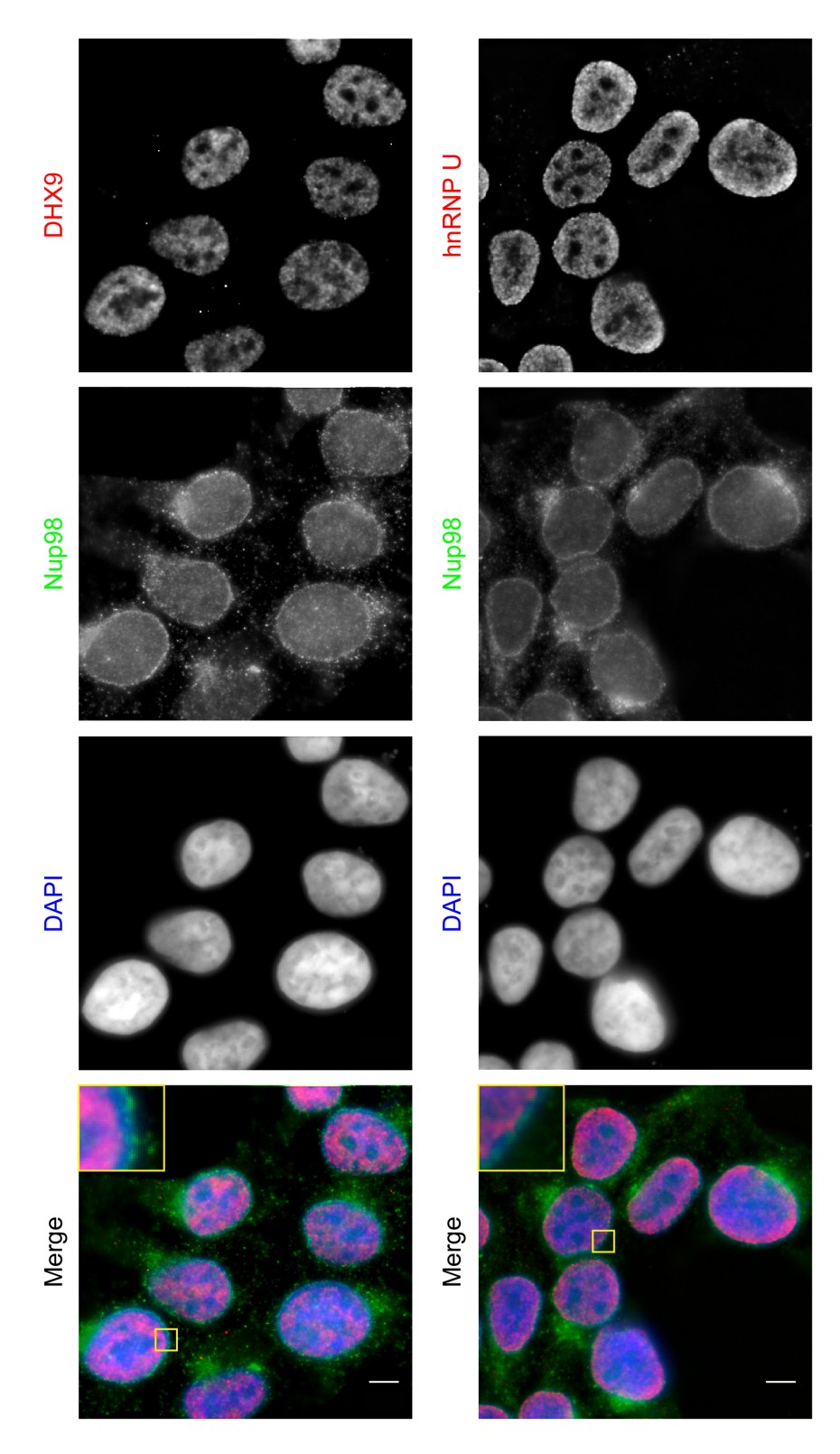

**Figure 3.** Localization of Nup98 with DHX9 and hnRNP U. The cellular distribution of Nup98, DHX9, and hnRNP U in HEK293T cells was examined by indirect immunofluorescence using antibodies directed against each protein as indicated. The positions of nuclei were determined using the DNA stain DAPI. Merged images showing DHX9 or hnRNP U (red), Nup98 (green), and DAPI-stained DNA (blue) are shown. Note, DHX9 and hnRNP U are partially excluded from the nucleoli, which exhibits reduced DAPI staining. Scale bars, 5 μm.

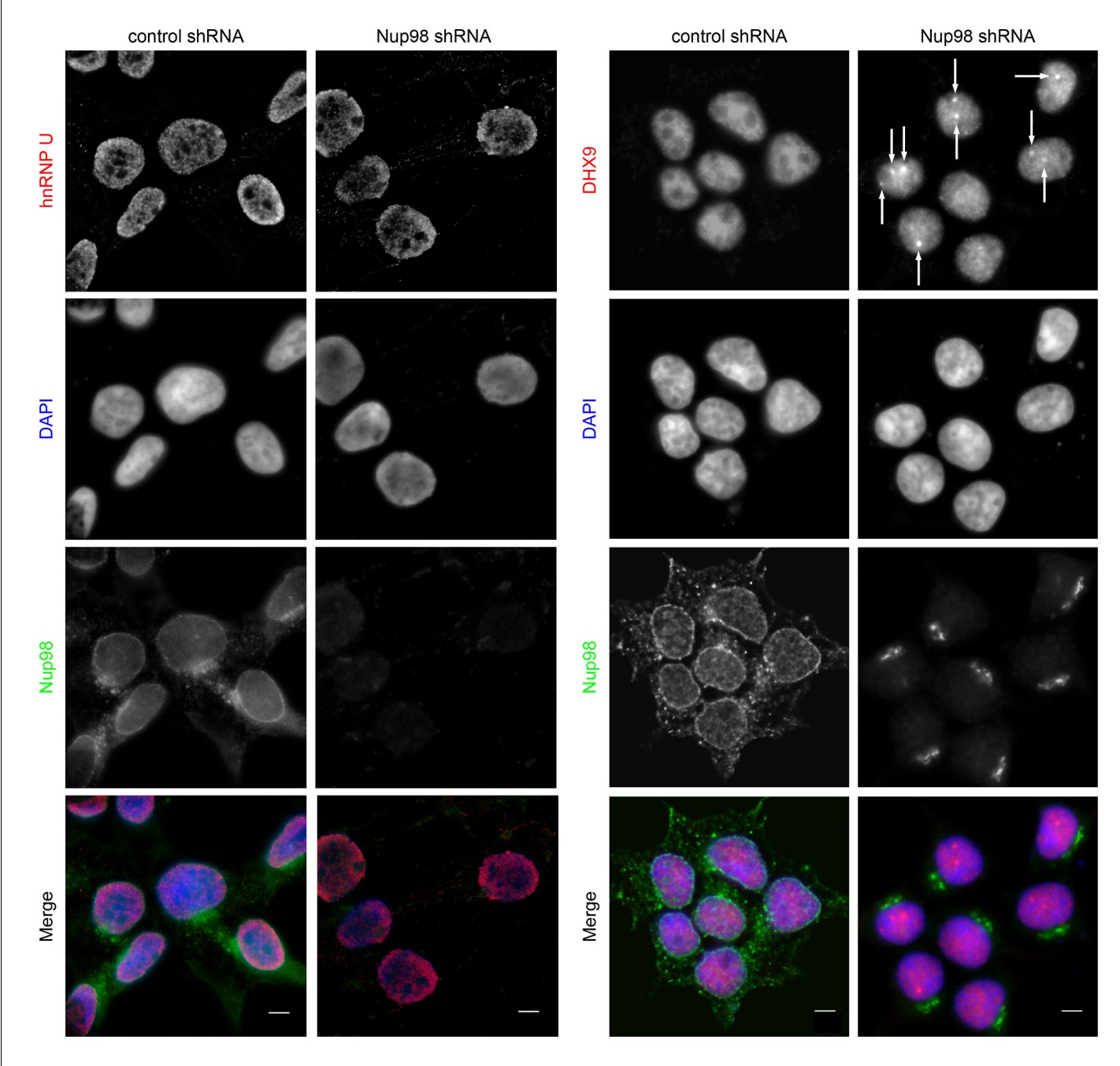

**Figure 4.** Nup98 depletion alters the intranuclear distribution of DHX9, but not hnRNP U. HEK293T cells were transfected with shRNA targeting Nup98 or with a control shRNA. Four days later the cellular distributions of Nup98 and either DHX9 or hnRNP U were examined by indirect immunofluorescence. Cells depleted of Nup98 show partial relocation of DHX9 into intranuclear foci (white arrows). Merged images show DHX9 or hnRNP U (red), Nup98 (green), and DAPI-stained DNA (blue). Scale bars, 5 μm. The cellular localization of Nup98 is not affected by depletion of DHX9 (see *Figure 4—figure supplement 1*). Protein depletion in these experiments was confirmed by immunoblot analysis (*Figure 4—figure supplement 2*).

The following figure supplements are available for figure 4:

**Figure supplement 1.** DHX9 depletion does not alter Nup98 localization in the cell.

**Figure supplement 2.** Immunoblotting of cell extracts following shRNA-mediated protein depletion.

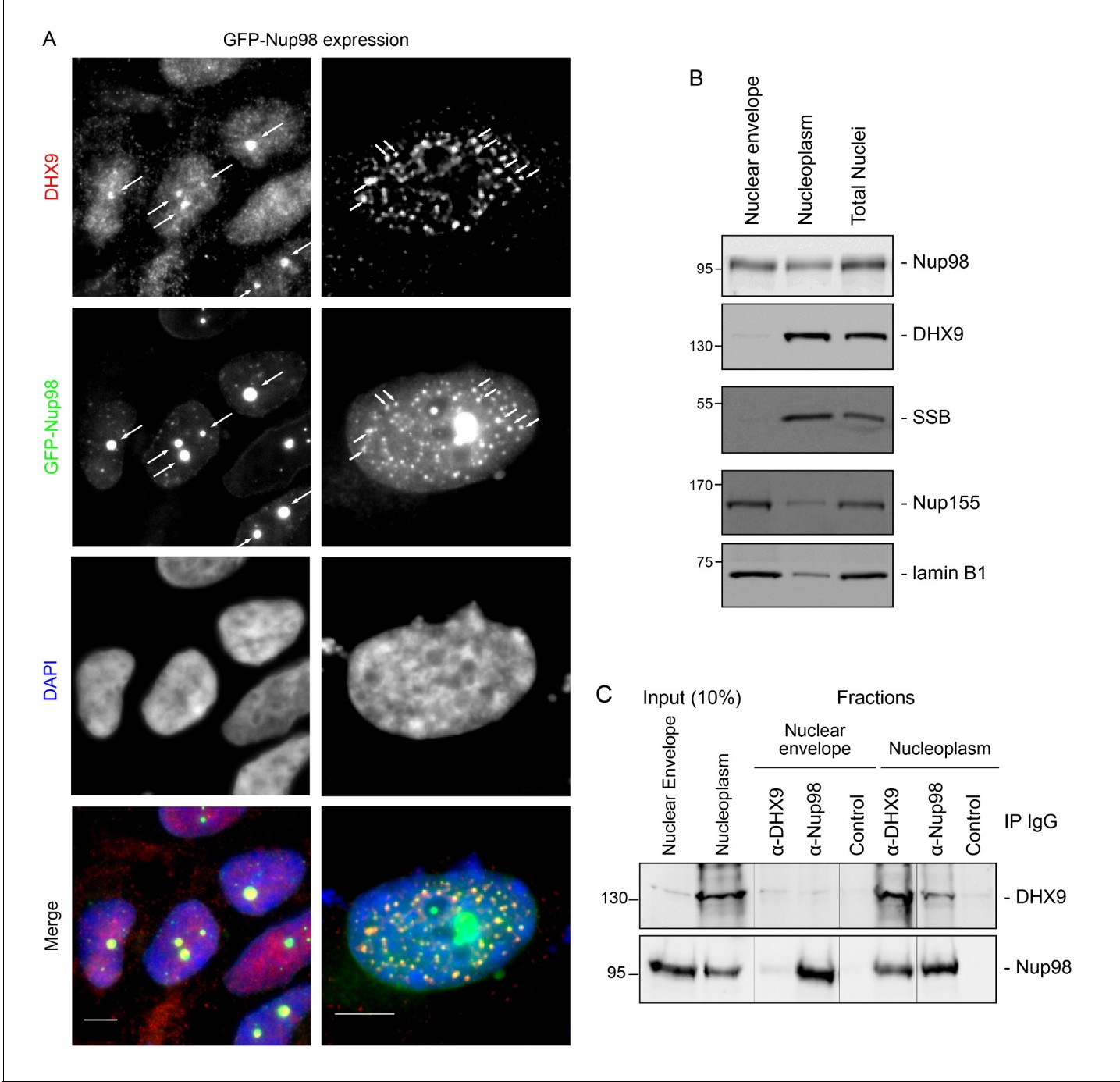

**Figure 5.** DHX9 interacts with intranuclear Nup98. (**A**) HEK293T cells expressing *GFP-NUP98* were used to compare DHX9 and GFP-Nup98 localization by immunofluorescence microscopy. Two magnifications are shown, each showing that upon *GFP-NUP98* expression intranuclear GFP-Nup98-containing foci form that contain DHX9. Examples of GFP-Nup98 colocalization with DHX9 are marked by arrows. Cells expressing higher levels of *GFP-NUP98* (right column) contain greater numbers of GFP-Nup98 foci and display even more pronounced DHX9 colocalization. Merged images show DHX9 (red), GFP-Nup98 (green), and DAPI-stained DNA (blue). Scale bars, 5 µm. In contrast, expression of *GFP* had no effect on DHX9 localization, and expressing *GFP* or *GFP-NUP98* had no impact on hnRNP U localization (*Figure 5—figure supplement 1*). The presence of GFP-Nup98 was confirmed by immunoblot (*Figure 5—figure supplement 2*). (**B**) HeLa cell nuclei were fractionated to produce nucleoplasmic and nuclear envelope fractions. Fractions were analyzed by western blotting using antibodies directed against the indicated proteins (right). The positions of molecular mass markers (shown in kDa) are indicated on the left. The fractionation procedure was evaluated by western blotting using antibodies directed against NE (lamin B and Nup155) and nucleoplasmic (SSB) proteins. (**C**) Nup98 or DHX9 were immunoprecipitated from nucleoplasmic and NE fractions derived from HeLa

*Figure 5 continued on next page*

*Figure 5 continued*

cell nuclei. Co-purifying proteins from the samples indicated above the panels were separated by SDS-PAGE and analyzed by immunoblotting to detect DHX9 and Nup98 as specified to the right of the panels. The positions of molecular mass markers (shown in kDa) are indicated on the left.

The following figure supplements are available for figure 5:

**Figure supplement 1.** GFP expression does not alter the localization of DHX9, nor does GFP-Nup98 alter hnRNP U localization.

**Figure supplement 2.** GFP or GFP-Nup98 expression does not alter cellular levels of DHX9 or hnRNP U.

**Figure supplement 3.** DHX9 is recruited to intranuclear GFP-Nup98$^{1-497}$ foci.

## Nup98 and DHX9 interact with a shared subset of mRNAs and gene loci

Given that Nup98 and DHX9 exist in a complex in vivo, and that Nup98 stimulates DHX9 activity in the presence of RNA, we would expect that Nup98 and DHX9 interact with a shared set of mRNAs. To assess this, we compared recently published mRNA binding datasets for Nup98 (*G Hendrickson et al., 2016*) and DHX9 (*Erkizan et al., 2015*). We find a statistically significant overlap in these datasets with ~37% of the Nup98-interacting mRNAs also detected bound to DHX9 and ~40% of the DHX9 bound transcripts interacting with Nup98 (p=2.5×10$^{-93}$; see Materials and methods). To directly test whether these proteins bind similar mRNAs, we immunoprecipitated DHX9 and Nup98 from cell lysates following crosslinking. By using stringent conditions that disrupt the DHX9-Nup98 interaction (*Figure 8—figure supplement 1*), we could assess the ability of each protein to bind RNA independent of one another and determine whether they interact with similar RNA species. RT-PCR was used to test whether specific mRNA species were associated with the immunopurified proteins. We tested for the presence of several potential interacting mRNAs, encoding JunD, Myc, FoxP2, HoxA2, and ZFY (*Erkizan et al., 2015*; *G Hendrickson et al., 2016*; *Hartman et al., 2006*; *Ranji et al., 2011*; *Weidensdorfer et al., 2009*; *Yugami et al., 2007*), and two predicted negative controls NHLH2, and HEXIM1. As anticipated, JunD, Myc, FoxP2, HoxA2, and ZFY encoding mRNAs were detected bound to both DHX9 and Nup98 (*Figure 8*). By contrast, both NHLH2 and HEXIM1, showed no interaction with either DHX9 or Nup98. Both Nup98 and DHX9 also interacted with the Adenovirus early region 1A (E1A) encoding RNA, a well known splicing reporter whose metabolism is regulated by several hnRNPs and RNA helicases (*Zheng, 2010*). These results suggest that Nup98 and DHX9 interact with, and potentially regulate, a shared set of mRNAs in vivo.

To investigate the interdependencies of mRNA-binding, we depleted either Nup98 or DHX9 and evaluated mRNA binding by the other factor. Note that depletion of Nup98 or DHX9 did not alter cellular levels of the other protein or the efficiency of immunoprecipitation (*Figure 9—figure supplement 1*). As shown in *Figure 9*, upon depletion of DHX9, five of six mRNAs (*E1A, FOXP2, HOXA2, MYC* and *ZFY*) showed a significant decrease in Nup98 association relative to the input as compared to mock-depleted cells. By contrast, depletion of Nup98 led to a significant increase in the amount of each of the six mRNAs bound to DHX9. The changes in the association of Nup98 or DHX9 with these mRNAs does not appear to be due to a change in the nuclear export status of the tested mRNAs (*Figure 9—figure supplement 2*). These results show that Nup98 and DHX9 influence each others association with mRNA, and are consistent with a model in which DHX9 promotes the association of Nup98 with specific mRNAs, and Nup98 facilitates the release of these mRNAs from DHX9.

To further evaluate the nature of the shared binding of Nup98 and DHX9 to this set of mRNAs, we used the DamID assay (*Franks et al., 2016*; *Vogel et al., 2007*) to determine whether the Nup98-DHX9 complex interacted with the gene loci encoding these mRNAs. For this analysis, genes encoding Nup98 or DHX9 fused to *E.coli* DNA methyltransferase (Dam) were introduced in to HEK293T cells. Modified DNA was then amplified, purified, and used in qPCR reactions to assess whether specific regions of the genome were bound to Nup98 and DHX9. As shown in *Figure 10*, both Nup98 and DHX9 mapped to the six gene loci whose transcripts were bound to Nup98 and DHX9. For *JUND* and *MYC*, two regions within these genes were examined: a 5' region containing the promoter and a region within the 3' half of the ORF. For both Nup98 and DHX9, robust binding

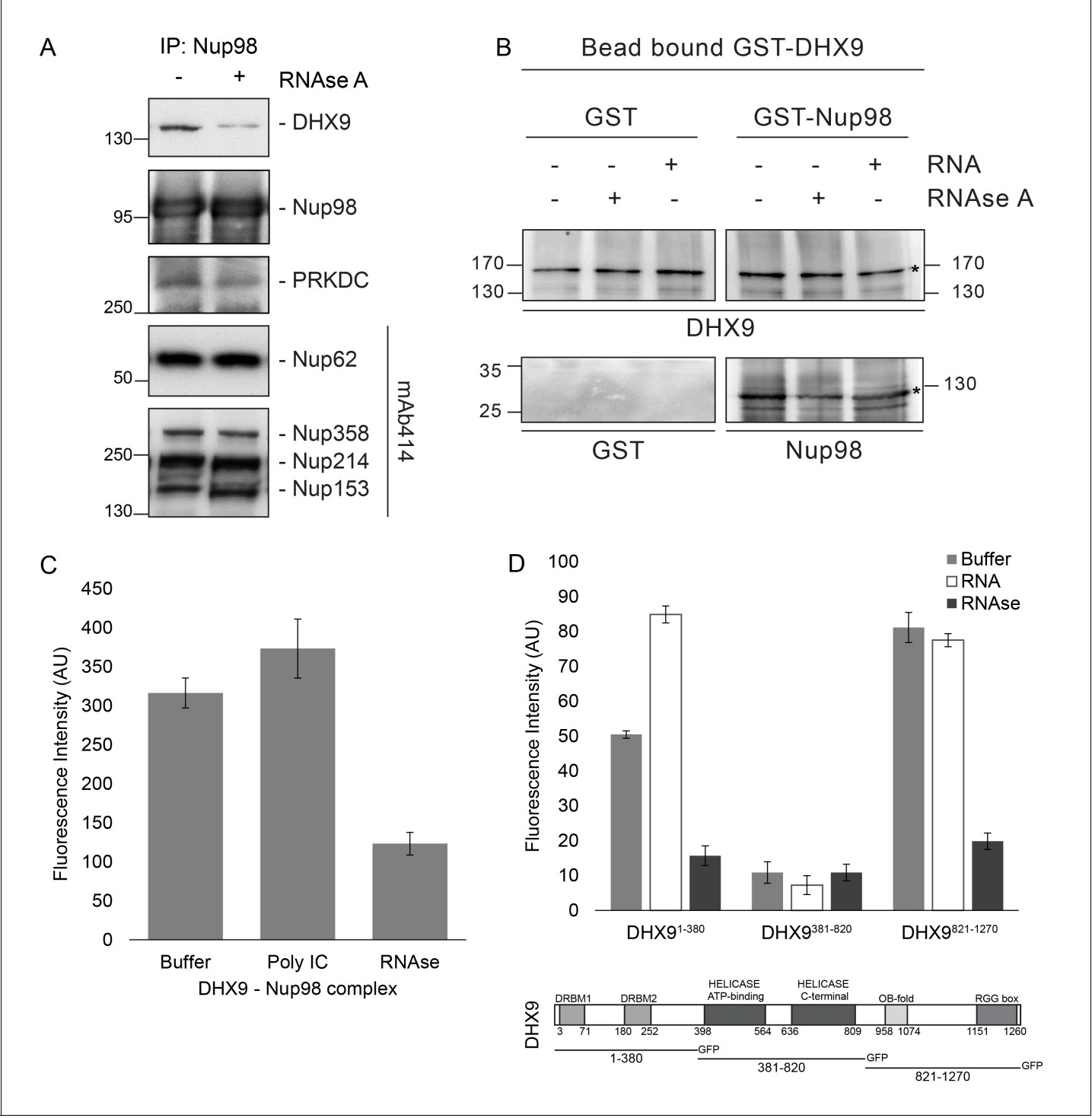

**Figure 6.** Nup98 binds directly to DHX9. (**A**) Nup98 was affinity purified from HEK293T cell lysates. Bead-bound protein complexes were then incubated with or without RNase A, and proteins remaining bound to Nup98 were analyzed by western blotting using antibodies directed against the indicated proteins (right). The positions of molecular mass markers (shown in kDa) are indicated on the left. (**B**) Anti-DHX9 antibodies coupled to beads were used to immobilize GST-DHX9. Bead-bound GST-DHX9 was incubated with GST-Nup98 or GST alone in the presence of RNA (poly I:C), RNase A, or buffer alone. Bound proteins were analyzed by western blotting using the indicated antibodies (below each panel). The top row of images shows the GST-DHX9 bait bound to beads. The bottom row of images shows GST and GST-Nup98 that bound to GST-DHX9 under the indicated conditions. Asterisks denote positions of GST-DHX9 and GST-Nup98. The positions of molecular mass markers (shown in kDa) are indicated. A similar interaction between untagged recombinant DHX9 and Nup98 was also detected (***Figure 6—figure supplement 1A***). (**C**) Bead halo assays were performed using

*Figure 6 continued on next page*

*Figure 6 continued*

DHX9 immobilized on beads with an anti-DHX9 antibody as bait and Nup98 as prey. Prior to the binding step, both proteins were incubated with RNase A, RNA (poly I:C), or buffer alone. Interactions of Nup98 with bead-bound DHX9 were detected by fluorescence microscopy with rabbit anti-Nup98 antibodies and Alexa Fluor 488 donkey anti-rabbit antibodies. Examples of images of beads used to quantify binding can be seen in *Figure 6—figure supplement 1B*. (D) Bead halo assays were performed using bead-bound GST-Nup98 (bait) and different domains of DHX9-GFP (prey; see bottom diagram). Proteins were incubated with RNA (poly I:C), RNase A, or buffer alone before binding. The interaction of bead-bound GST-Nup98 with DHX9-GFP domains was detected by fluorescence microscopy. Examples of images of beads can be seen in *Figure 6—figure supplement 1C*. For C and D, bead halo assays were performed with purified recombinant proteins. Plots show average fluorescence intensity values of beads (arbitrary units) corrected against negative control assays (see Materials and methods). Results from three biological replicates are shown. Error bars indicate standard deviation between biological replicates.

The following figure supplement is available for figure 6:

**Figure supplement 1.** In vitro interaction of Nup98 and DHX9.

was detected to the 5' promoter regions of these genes, while binding to regions within their ORFs were lower or absent. Similarly, no detectable binding was observed to the NHLH2 and HEXIM1 genes, consistent with our observation that the transcripts from these genes were not detected in association with Nup98 and DHX9 (*Figure 10*). Of note, DamID experiments performed with a fusion (Dam-Nup98$^{1–504}$) containing only the N-terminal FG/GLFG domain of Nup98 shown to be sufficient for DHX9 binding (*Figure 5—figure supplement 3*) displayed a similar chromatin-binding profile (*Figure 10—figure supplement 1*). The binding of Nup98 and DHX9 to the gene loci tested was also interdependent on one another. Depletion of Nup98 or DHX9 significantly reduced the interactions of its binding partner with the target gene (*Figure 10* and *Figure 10—figure supplement 1*). We therefore suggest that the Nup98-DHX9 complex binds to specific genes and their transcripts.

## Nup98 stimulates DHX9-mediated transcription

DHX9 and Nup98 have been linked to various steps in mRNA metabolism, including transcription (*Fidaleo et al., 2016*; *Franks et al., 2016*; *Lee and Pelletier, 2016*; *Liang et al., 2013*; *Light et al., 2013*; *Pascual-Garcia et al., 2014*). Consistent with these data, the specific genes we detected bound to Nup98 and DHX9 exhibited altered transcript levels upon depletion of these proteins (*Figure 11*). Furthermore, analysis of RNA-Seq data (*Chen et al., 2014*; *Franks et al., 2016*) revealed shared sets of genes with altered transcription upon depletion of these proteins. A comparison of these data sets shows significant overlap in the identity of gene products affected by the depletion of either protein (287 genes with altered expression upon DHX9 or Nup98 depletion, p=3.24×10$^{-36}$; see Material and methods), consistent with the idea that these proteins form a functional complex. Interestingly, a significant number (p-value $2.38 \times 10^{-4}$) of those genes exhibiting altered expression upon Nup98 depletion contain a putative cAMP-response element (CRE) (*Zhang et al., 2005*), a regulatory element whose transcriptional activity can be regulated by DHX9 (*Aratani et al., 2001*; *Fidaleo et al., 2016*; *Lee and Pelletier, 2016*).

To more directly assess the functional role of Nup98 in DHX9-mediated transcription, we used a CRE-luciferase reporter assay. This assay has been used to evaluate the role of DHX9 in transcription, including defining the contributions of its ATPase activity to its role in transcription (*Aratani et al., 2001*). Similar to previous reports, expression of exogenous DHX9 in cells containing the CRE-luciferase reporter increased production of luciferase (*Figure 12*). Point mutants in DHX9 that reduce (DHX9$^{I347A}$) or eliminate (DHX9$^{K417R}$) ATPase activity show reduced stimulation of reporter expression (*Aratani et al., 2001*). Since our in vitro assays showed that Nup98 could stimulate the ATPase activity of DHX9, we tested whether overexpression of Nup98 could stimulate the DHX9-mediated expression of CRE-luciferase. In the absence of exogenous DHX9, expression of Nup98 had no significant effect on the expression of luciferase. However, Nup98 expression stimulated luciferase production in the presence of DHX9 or the ATPase compromised DHX9$^{I347A}$ mutant, while having no significant stimulatory impact on luciferase expression in the presence of the ATPase dead mutant (DHX9$^{K417R}$). These results are consistent with our in vitro observations showing Nup98 can stimulate the ATPase activity of DHX9, and they suggest that the stimulatory effect of Nup98 binding to DHX9 supports its role in transcription.

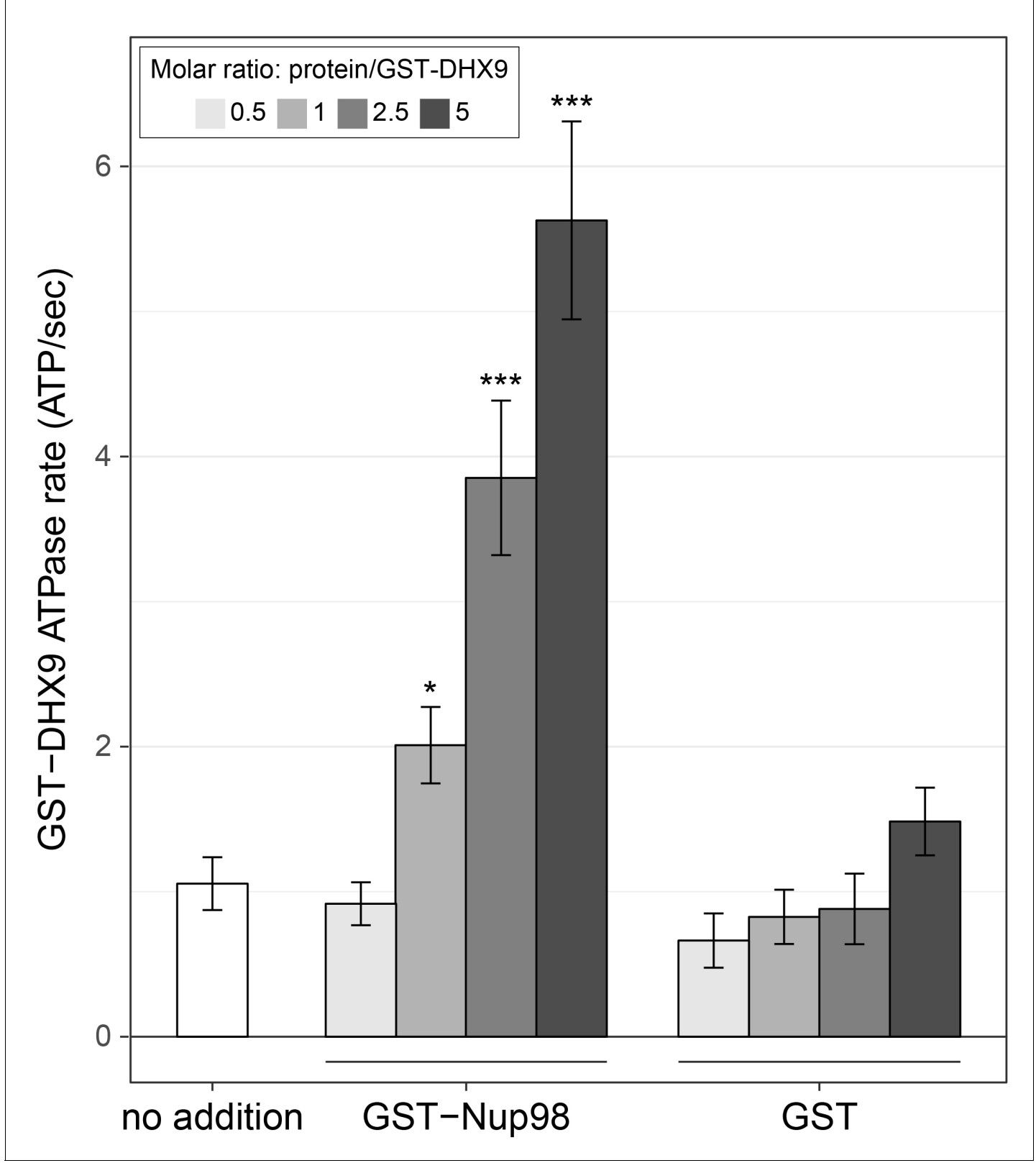

**Figure 7.** Nup98 stimulates DHX9 ATPase activity. The ATPase activity (ATP hydrolysis rate) of purified recombinant GST-DHX9 in the presence of RNA alone (no addition) or following the addition of increasing concentrations of GST-Nup98 or GST (show as the molar ratio of the added protein to that of GST-DHX9) is shown on the y-axis. Error bars indicate standard deviation. Results from three biological replicates were submitted to ANOVA followed by Tukey HSD tests (*** indicates adjusted p-values < 0.001 and * indicates adjusted p-values < 0.05 for Tukey HSD in pairwise comparison between a

*Figure 7 continued on next page*

*Figure 7 continued*

reaction containing GST-Nup98 and a reaction containing GST in similar molar amounts). Similar results were obtained using untagged versions of DHX9 and Nup98 (*Figure 7—figure supplement 1B*). The addition of GST-Nup98 constructs or GST alone had no effect on the ATPase rate of GST-DHX9 in the absence of RNA (*Figure 7—figure supplement 1C*).

The following figure supplement is available for figure 7:

**Figure supplement 1.** DHX9 ATPase assay.

Steps in mRNA metabolism are often tightly coupled, including transcription and mRNA splicing (*Alpert et al., 2017*; *Saldi et al., 2016*). Among the curated DHX9 protein-protein interactions (*Figure 2A*), factors functioning in mRNA splicing are among the most abundant. In addition, DHX9 has been implicated in splicing regulation (*Bratt and Ohman, 2003*; *Selvanathan et al., 2015*), raising the possibility that the interactions of Nup98 and DHX9 may also play a role in this process. Data sets from RNA-Seq analysis of Nup98 and DHX9 depleted cells reveal a significant overlap in gene products exhibiting altered splicing upon depletion of each protein (see Material and methods). DHX9 depletion altered the splicing of 866 genes, of these 217 genes also show altered splicing upon Nup98 depletion (p=$2.03 \times 10^{-43}$). Based on this information, we examined splicing isoforms of the well-characterized E1A mRNA, which interacted with DHX9 and Nup98 (*Figure 8*). Different E1A splicing intermediates (13S, 12S, 11S, 10S and 9S) have been characterized (*Stephens and Harlow, 1987*), the abundance of which could be quantified following depletion of Nup98 or DHX9 (*Figure 13*). Depletion of Nup98 or DHX9 resulted in a 1.9 or 1.8 fold increase in pre-spliced isoform of the transcript (*Figure 11*). Furthermore, we observed differential effects on the levels of the various splicing isoforms. Depletion of Nup98 led to significantly increased levels of the 12S, 11S and 10S isoforms. A similar increase in 12S and 11S isoform was detected in cells depleted of DHX9. DHX9 depletion also caused significant decreases in 9S and 13S abundance. These results suggest that the interactions of Nup98 and DHX9 with specific mRNAs, such as E1A, regulates their splicing.

## Discussion

Several previous publications have established the importance of Nup98 in regulating gene expression (*Kalverda et al., 2010*; *Liang et al., 2013*; *Light et al., 2013*; *Pascual-Garcia et al., 2014*; *Singer et al., 2012*). However, our understanding of the mechanistic role of Nup98 in this process has lagged due to our limited knowledge of Nup98 binding partners, most notably in the nucleoplasm, and the consequences of these interactions on the functions of the interacting partners. In this work, we focused on a possible mechanism by which Nup98 can alter gene expression through its interaction with, and regulation of, the RNA helicase DHX9. We have shown using a combination of in vitro and in vivo assays that Nup98 directly binds DHX9 in the nucleoplasm and this interaction is facilitated by RNA. Importantly, binding of Nup98 to DHX9 can stimulate the ATPase activity of DHX9 and support the role of this DExH/D-box protein in the transcription and splicing of a subset of genes. Consistent with these observations, our analysis revealed that Nup98 and DHX9 bind to similar gene loci and mRNAs, and these interactions are interdependent upon one another. In aggregate, our observations lead us to conclude that intranuclear Nup98 functions as a regulator of DHX9.

### The interaction of Nup98 with DHX9

Immunopurified Nup98 revealed associated proteins with known roles in mRNA metabolism (*Figures 1* and *2*), mainly hnRNP proteins and RNA helicases (*Han et al., 2010*, *Bourgeois et al., 2016*) suggesting a functional link between Nup98 and mRNA metabolism. We envisage that many of these proteins are components of Nup98-interacting protein complexes, but most are unlikely to bind directly to Nup98. As others have concluded from MS data (*Cox and Mann, 2011*; *Liu et al., 2004*; *Mazloom et al., 2011*), we speculated that proteins directly interacting with Nup98 were more likely to be among those species most highly represented by unique peptides in our MS analysis, which led us to focus on DHX9. This seemed reasonable as previous studies have described interactions between RNA helicases and nucleoporins, including the interactions of yeast Dbp5 with

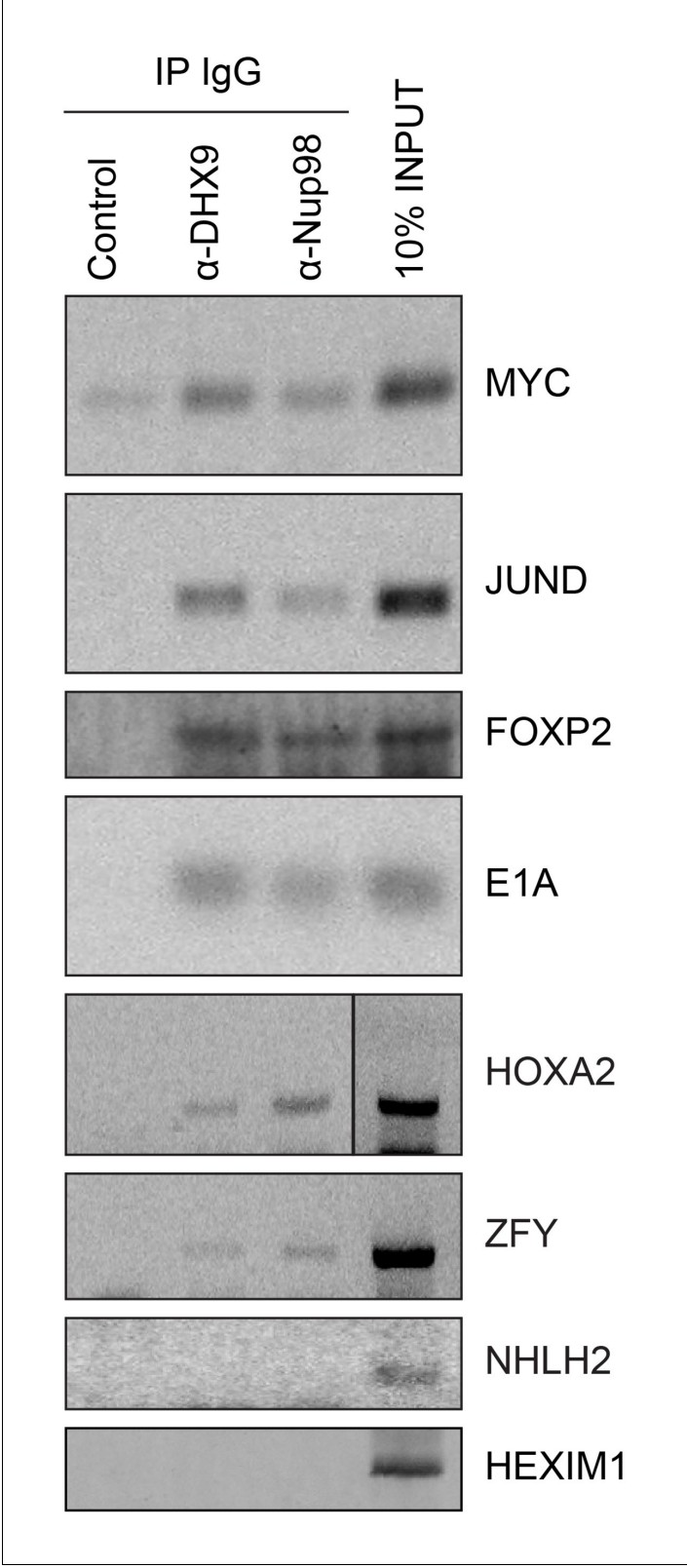

**Figure 8.** Nup98 directly interacts with target mRNA molecules. Following crosslinking of HEK293T cells to preserve protein/RNA complexes, cell lysates were incubated with beads coupled to a control IgG (α-GFP) or beads coupled to Nup98 or DHX9 specific antibodies. RNA present in immunoprecipitated complexes and total

*Figure 8 continued on next page*

*Figure 8 continued*
cellular RNA (10% input) was used as template in RT-PCR reactions containing primers (*Supplementary file 1D*) specific to regions of several cDNAs whose genes are denoted on the right.
The following figure supplement is available for figure 8:

**Figure supplement 1.** Analysis of protein immunoprecipitation.

Nup159 and human DDX19 with Nup214, and the role of these interactions in modulating ATP-dependent helicase activity and mRNA export (*Montpetit et al., 2011*; *Napetschnig et al., 2009*; *Noble et al., 2011*; *von Moeller et al., 2009*; *Weirich et al., 2004*).

DHX9, also termed RNA helicase A (RHA), is a member of the helicase superfamily 2. Most of the proteins known to function as RNA chaperones or RNA-protein complex remodelers in this super-family are found within the DEAD-box (DDX) and the DEAH/RHA (DHX) families (*Jarmoskaite and Russell, 2014*). Both DHX and DDX helicases contain a highly conserved helicase domain that mediates nucleotide binding and hydrolysis and is linked to binding of nucleic acid (*Jarmoskaite and Russell, 2014*; *Stevens, 2010*). Flanking the helicase domain, DHX family members possess variable N- and C-terminal domains that, while often containing shared sequence motifs, contribute to the diverse functions of family members (*Jankowsky and Fairman, 2007*; *Linder and Jankowsky, 2011*). DHX9 contains two double-stranded RNA binding domains (dsRBD) within the N-terminal third of the protein (*Zhang and Grosse, 1997*), while the C-terminal third of the protein contains an oligonucleotide/oligosaccharide (OB)-binding fold and an RGG-box, a domain that characteristically binds single-stranded nucleic acids. Several proteins have been shown to bind the N-terminus, C-terminus, or both regions of DHX9 (*Lee and Pelletier, 2016*).

The N- and C-terminal regions of DHX9 containing the dsRBDs and the RGG-box are thought to be spatially positioned in close proximity and contribute to the nucleic acid binding properties of DHX9 (*Zhang and Grosse, 1997*). As mentioned above, these regions also contribute binding surfaces for interacting proteins, and in this study we show that both N- and C-terminal regions of DHX9 bind the N-terminal FG/GLFG repeat region of Nup98 (*Figure 5—figure supplement 3* and *Figure 6D*). While the nature of these interactions remains to be examined in greater detail, studies of another DHX member, yeast Prp43, and its binding partner Ntr1 offer possible insights into the nature of the interactions between DHX9 and Nup98. Like DHX9, Prp43 contains an OB-binding fold, which binds to an intrinsically unstructured, N-terminal region of Ntr1 (*Christian et al., 2014*). Similarly, the C-terminal region of DHX9 contains an OB-binding fold that binds the unstructured FG/GLFG repeat region of Nup98 (*Figure 5—figure supplement 3* and *Figure 6D*). Intriguingly, the unstructured regions of Ntr1 contain a 'G-patch' motif rich in glycines and bulky, hydrophobic residues (*Aravind and Koonin, 1999*), a compositional property shared with the FG/GLFG repeats of Nup98. When Ntr1 binds to Prp43, the conformation imparted on the 'G-patch' motif facilitates its binding to RNA (*Christian et al., 2014*). We speculate that the binding of Nup98 to DHX9 may also impart structural features on the FG/GLFG repeat regions of Nup98 that facilitates binding to RNA. This idea is consistent with our observation that the binding of certain mRNAs to Nup98 is facilitated by DHX9 (*Figure 9*; see below).

Reciprocally, the binding of Nup98 to N- and C-terminal regions of DHX9 increases the ATPase activity of DHX9 (*Figure 7* and *Figure 7—figure supplement 1*). Other factors also function similarly to regulate DHX9. For example, Werner Syndrome helicase interacts with both N- and C-terminal regions of DHX9 to inhibit DHX9 activity (*Friedemann et al., 2005*). In another case, the catalytic subunit of DNA-dependent protein kinase (PRKDC) interacts with DHX9 and increases ATPase activity (*Mischo et al., 2005*). Mechanistically, the binding of Ntr1 to Prp43 again offers a precedent for how these binding events could regulate DHX9 activity (*Christian et al., 2014*; *Tanaka et al., 2007*).

## Nup98 and DHX9 interact in the nucleoplasm

Previous analyses of the subcellular distribution of DHX9 suggest it is largely restricted to the nucleo-plasm (*Zhang et al., 1995*), but it is excluded from nucleoli and shows no obvious accumulation at NPCs (*Figure 3*) (*Lee and Pelletier, 2016*; *Uhlén et al., 2015*; *Zhang et al., 1995*). Several

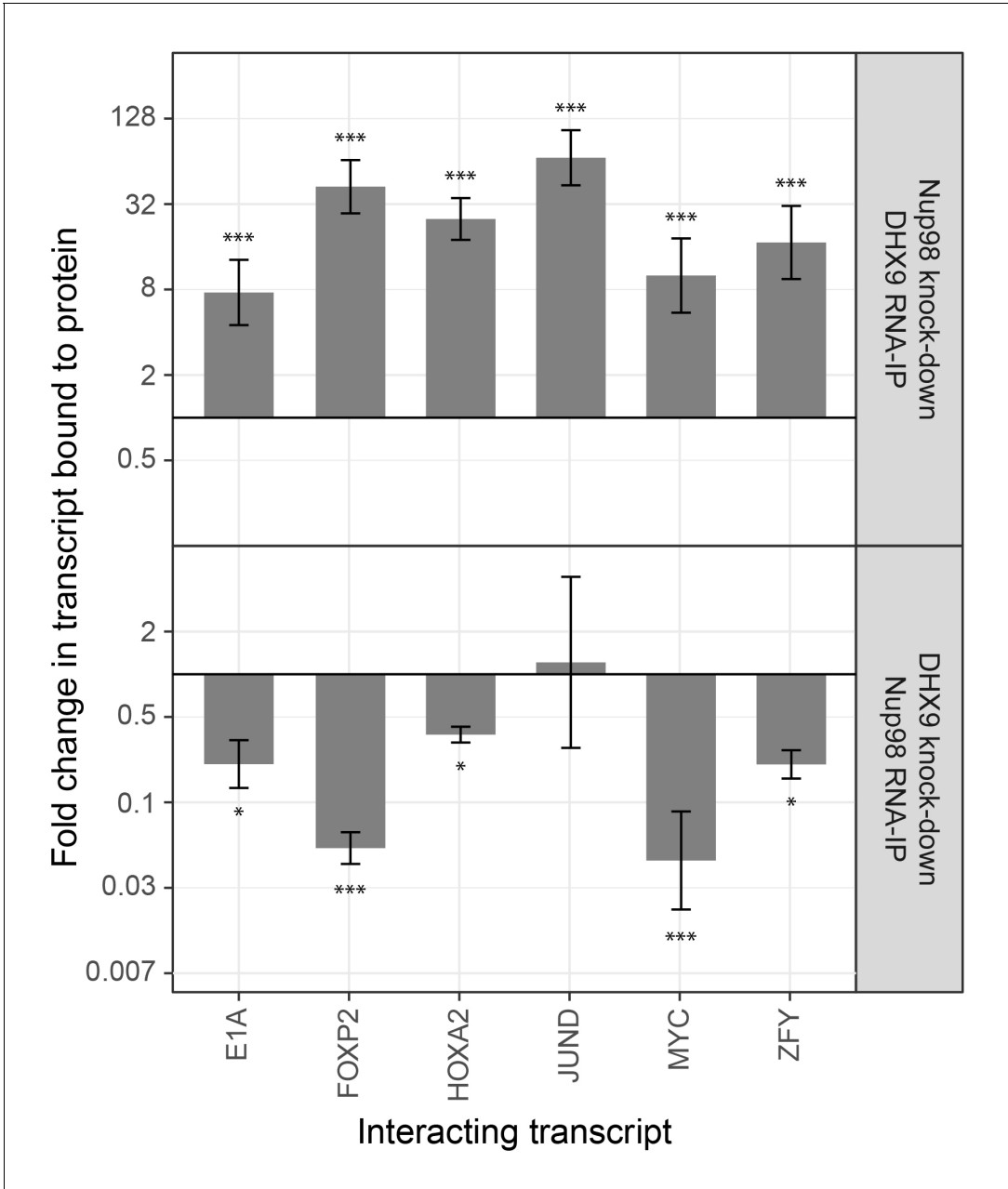

**Figure 9.** The association of Nup98 or DHX9 with specific mRNAs is altered by depletion of its binding partner. HEK293T cells were transfected with a control shRNA or an shRNA targeting Nup98 or DHX9. RNA immunopurified with Nup98 or DHX9 was reverse transcribed and used in qPCR reactions to assess the levels of indicated transcripts (x-axis). The ratio of bound mRNA relative to input was determined for each transcript listed. The fold-change in this ratio is relative to that determined from mock-depleted cells and is shown on the y-axis. The top panel shows the results of mRNA bound to DHX9 upon Nup98 depletion and the bottom panel mRNA bound to Nup98 upon DHX9 depletion. Error bars indicate standard deviation for biological replicates. Results from three biological replicates were submitted to ANOVA followed by Tukey HSD tests. The *** indicates adjusted p-values < 0.001 and * < 0.05 for Tukey HSD in pairwise comparisons between depleted and mock depleted samples.

The following figure supplements are available for figure 9:

**Figure supplement 1.** Analysis of protein immunoprecipitation from HEK293T cells depleted of Nup98 or DHX9.

**Figure supplement 2.** Nup98 or DHX9 depletion has no significant impact on the nuclear or cytoplasmic abundance of target mRNAs.

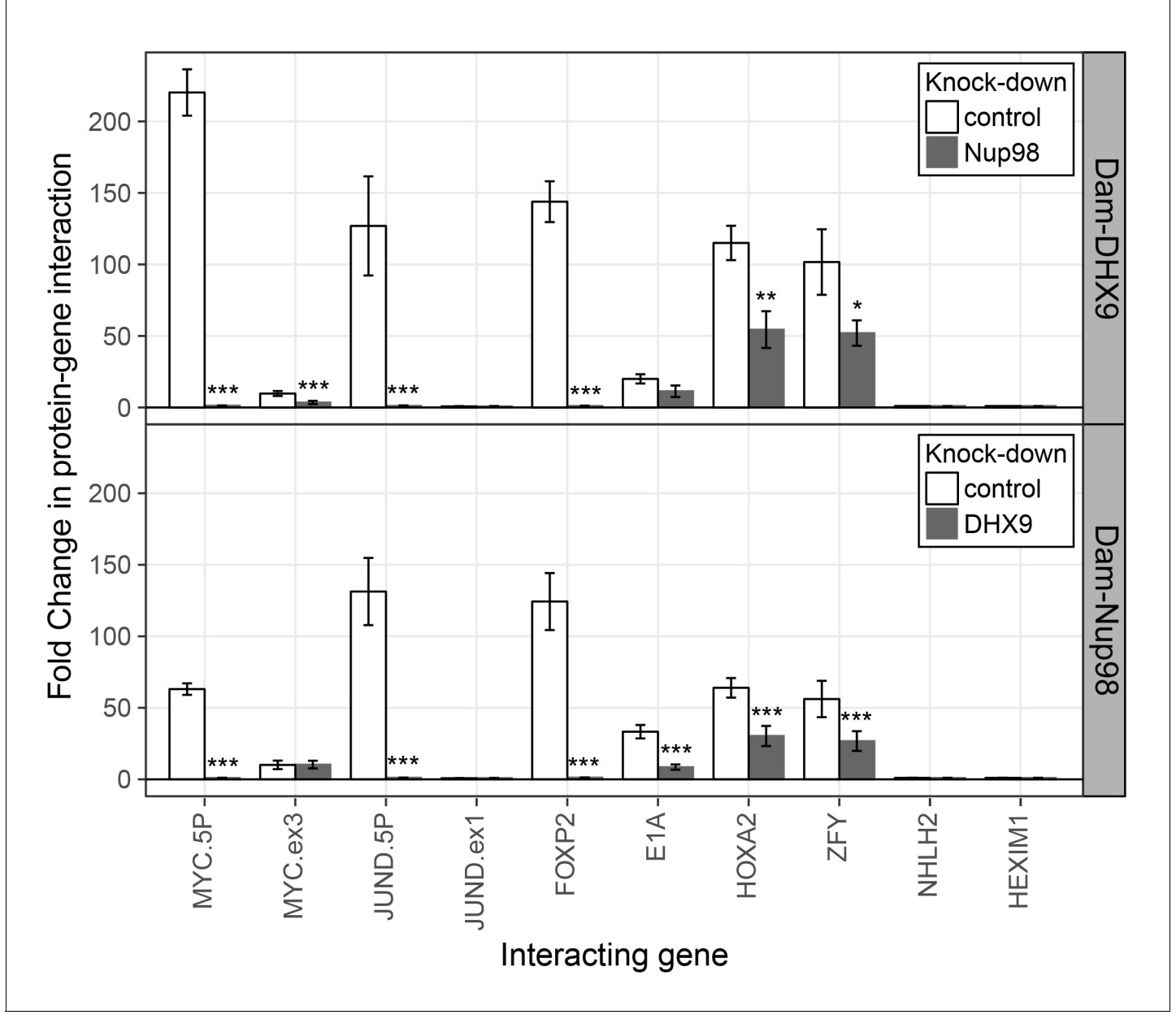

**Figure 10.** Nup98 and DHX9 associate with similar gene loci and their binding is interdependent. HEK293T cells stably expressing Dam-GFP, Dam-Nup98 or Dam-DHX9 were transduced with lentivirus encoding a control shRNA (white) or an shRNA targeting Nup98 or DHX9 (gray) and DamID analysis was performed. The association of Dam-Nup98 and Dam-DHX9 to the indicated gene loci is represented as the fold change (x-axis) relative to a Dam-GFP control. Error bars indicate standard deviation for biological replicates. For the top and bottom graphs, results from three biological replicates were submitted to two-way ANOVA followed by Tukey HSD tests. Adjusted p-values are indicated as *** < 0.001 < ** < 0.01 < * < 0.05 for Tukey HSD in pairwise comparisons between mock and Nup98 or DHX9 depleted cells for each gene tested.

The following figure supplement is available for figure 10:

**Figure supplement 1.** The N-terminal FG/GLFG region of Nup98 associates with specific gene loci and is altered by depletion of DHX9.

observations lead us to conclude that the Nup98-DHX9 complex also resides in the nucleoplasm. First, both depletion and overexpression of Nup98 altered the intranuclear distribution of DHX9, including the recruitment of DHX9 to intranuclear Nup98 foci (*Figure 5A*). Second, nuclear fractionation revealed that Nup98-DHX9 complexes are present primarily in the nucleoplasm (*Figure 5C*).

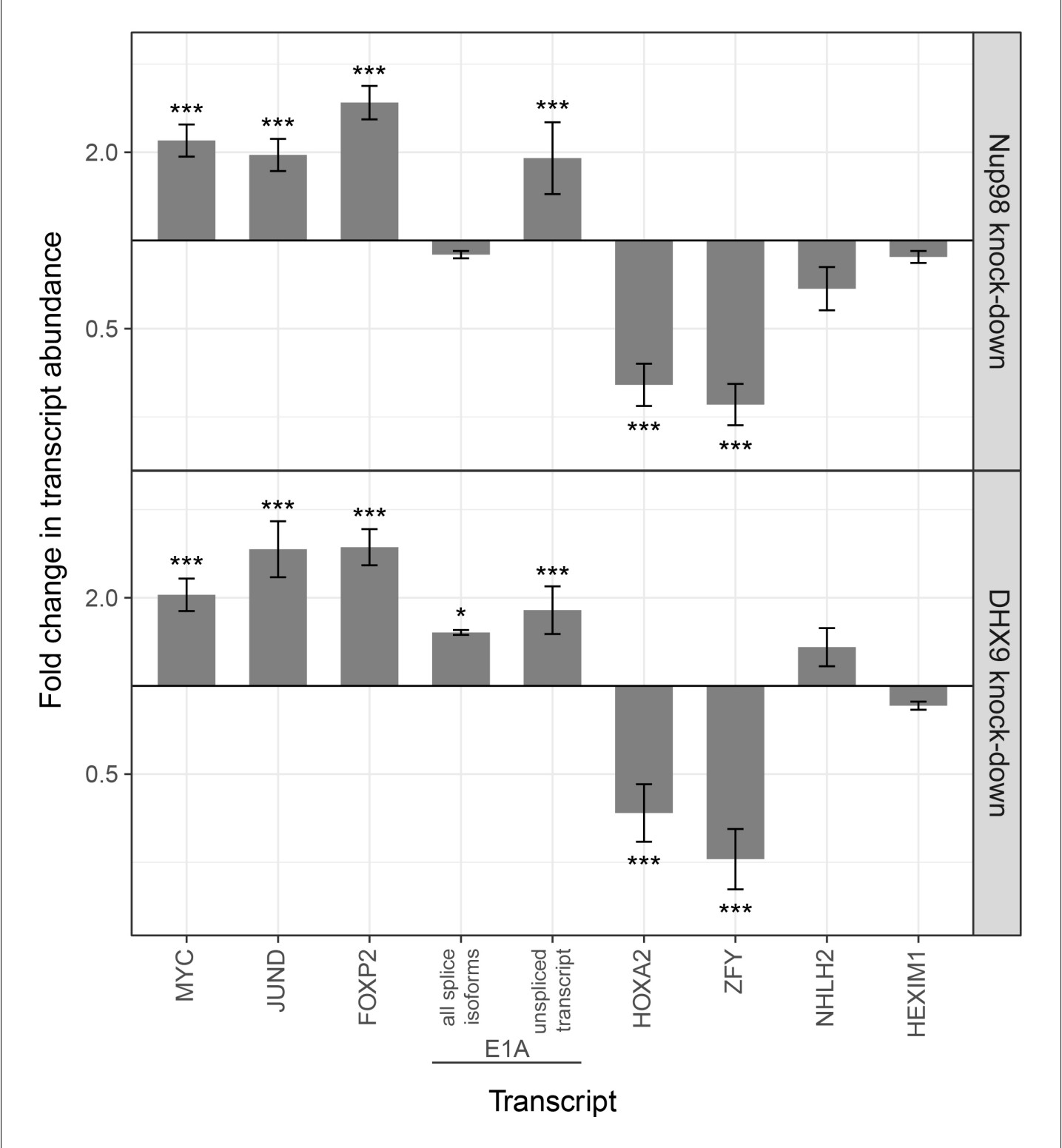

**Figure 11.** Nup98 or DHX9 depletion alters the abundance of target mRNAs. HEK293T cells were transduced with a control shRNA or an shRNA targeting Nup98 or DHX9. RNA was purified and transcript levels from the indicated genes (x-axis) were reverse transcribed and quantified by qPCR. Fold changes (y-axis) in the abundance of different transcripts upon Nup98 (top) or DHX9 (bottom) depletion relative to transcript abundance in mock depleted cells are shown. Error bars indicate standard deviation for biological replicates. Results from three biological replicates were submitted to

*Figure 11 continued on next page*

*Figure 11 continued*

ANOVA tests followed by Tukey HSD tests. p-values are indicated as *** < 0.001 and * < 0.05 for Tukey HSD in pairwise comparisons between mRNA levels from depleted and mock depleted cells.

Finally, DamID analysis established that the association of Nup98 and DHX9 with the same gene loci is interdependent on one another (*Figure 10*). Together these results strongly support the existence of an intranuclear Nup98-DHX9 complex. Given that DHX9 appears to be a more abundant protein than Nup98 (*Montague et al., 2015*; *Schaab et al., 2012*; *Wang et al., 2012*; *Wilhelm et al., 2014*), and that other DHX9 interacting partners appear to bind similar regions of DHX9

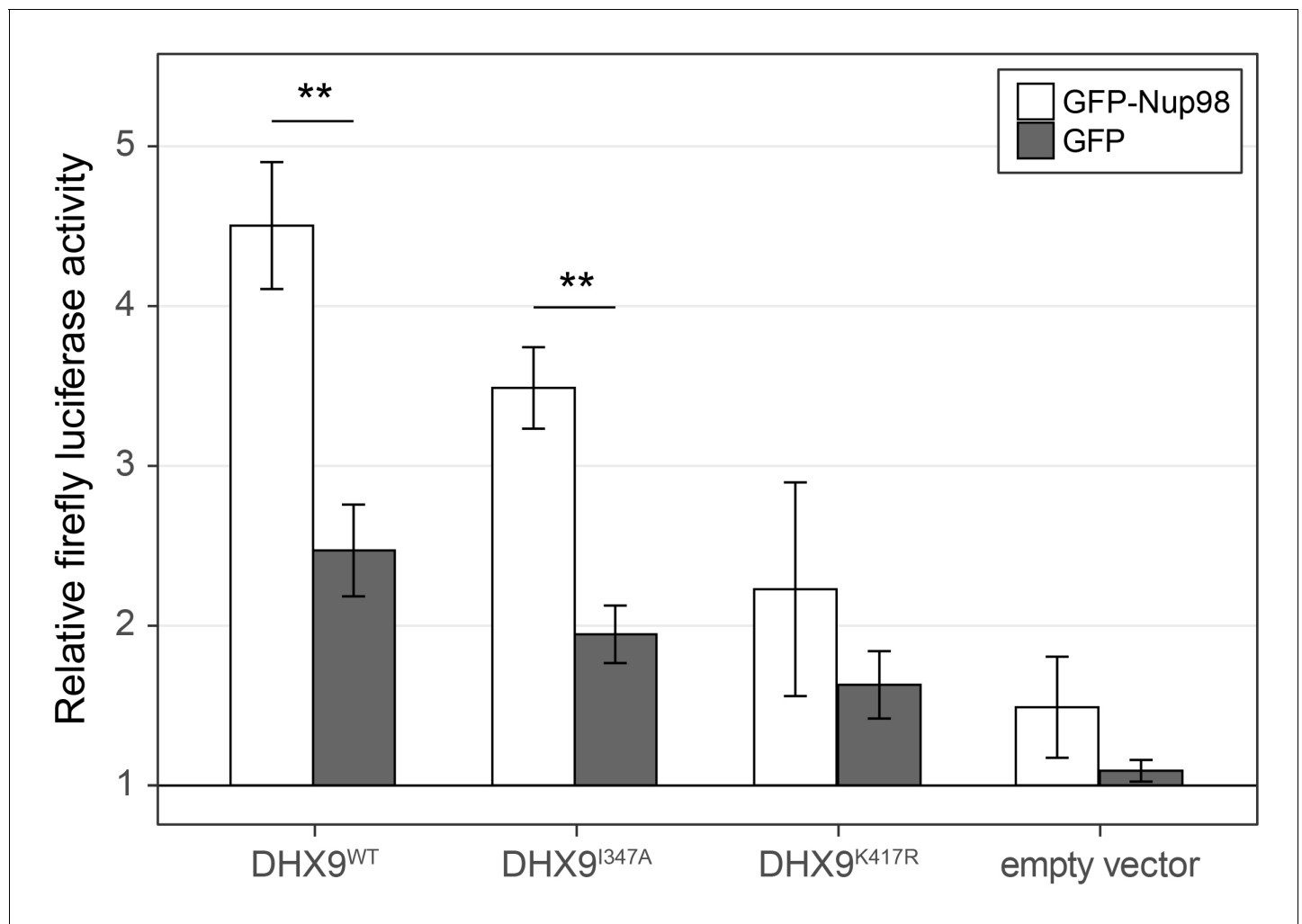

**Figure 12.** Nup98 stimulates the transcriptional activity of DHX9. HEK293T cells transfected with the luciferase gene under control of a cAMP-regulatory element (CRE) were co-transfected with two plasmids, one containing *GFP-NUP98* or *GFP* and another containing either $DHX9^{WT}$, the point mutant $DHX9^{I347A}$, the point mutant $DHX9^{K417R}$ or an empty plasmid. Luciferase activity is shown on the y-axis. The luciferase activity from cells transfected with luciferase plasmid alone was designated 1. Each value of relative luciferase activity represents the mean ± standard deviation (n = 3). (** indicate p-value < 0.01 in T-test comparing normalized luciferase activity in cells transfected with GFP-Nup98 versus GFP). DHX9 and the DHX9 point mutants are expressed at similar levels (*Figure 12—figure supplement 1*).

The following figure supplement is available for figure 12:

**Figure supplement 1.** *DHX9* point mutant constructs are expressed at levels similar to WT.

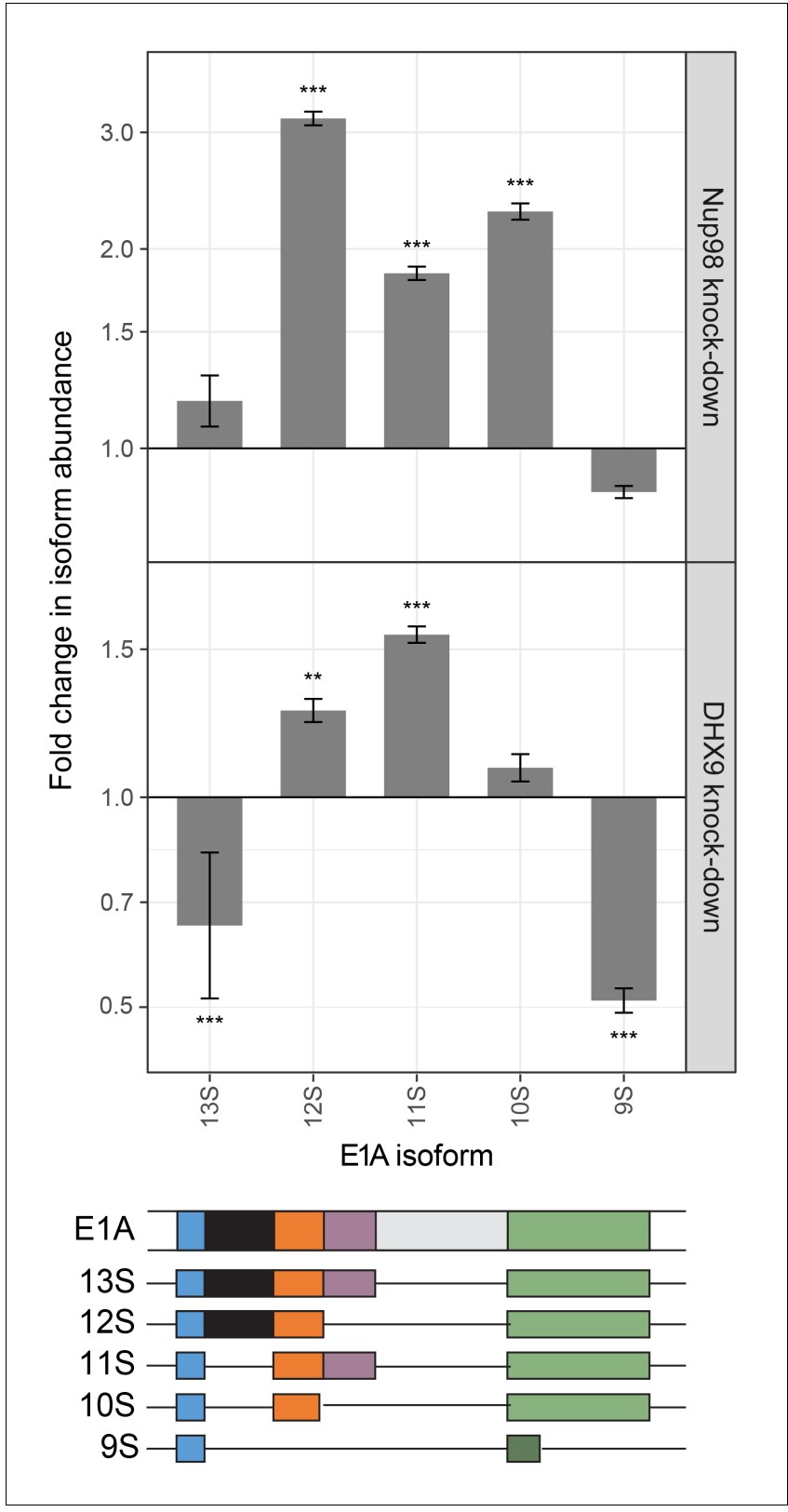

**Figure 13.** Nup98 or DHX9 depletion affects alternative splicing of E1A mRNA. HEK293T cells were transfected with a control shRNA or an shRNA targeting Nup98 or DHX9. RNA from these cells was reverse transcribed and cDNAs used as template in qPCR reactions containing primers specific to different splice isoforms of the E1A mRNA (see bottom diagram). The abundance of each E1A splice isoform was normalized to total E1A transcript

*Figure 13 continued on next page*

*Figure 13 continued*

present in the same sample. Fold change in the normalized abundance of different E1A splice isoforms between the knock-down and control samples are shown in the y-axis. Error bars indicate standard deviation for biological replicates. Results from three biological replicates were submitted to ANOVA followed by Tukey HSD tests. Adjusted p-values are indicated as *** < 0.001 < ** < 0.01 for Tukey HSD in pairwise comparisons between E1A splice isoforms mRNA amounts from cells depleted of DHX9 or Nup98 and mock-depleted cells.

(*Anderson et al., 1998*; *Erkizan et al., 2015*; *Jin et al., 2011*; *Nakajima et al., 1997*; *Pellizzoni et al., 2001b*; *Robb and Rana, 2007*; *Sadler et al., 2009*; *Smith et al., 2004*; *Tetsuka et al., 2004*), we assume that Nup98 binds and regulates a subpopulation of DHX9. Overall, the binding partners (e.g. Nup98, Werner Syndrome helicase, and PRKDC) of DHX9 are likely key to determining the localization of DHX9, the specific mRNAs it binds, and its overall involvement in gene expression (*Anderson et al., 1998*; *Erkizan et al., 2015*; *Jin et al., 2011*; *Nakajima et al., 1997*; *Pellizzoni et al., 2001b*; *Robb and Rana, 2007*; *Sadler et al., 2009*; *Smith et al., 2004*; *Tetsuka et al., 2004*).

## Nup98-DHX9 complex functions to regulate transcription and mRNA processing

Nup98 and DHX9 have each independently been shown to play a role in regulating gene transcription (*Capelson et al., 2010*; *Kalverda and Fornerod, 2010*; *Lee and Pelletier, 2016*; *Liang et al., 2013*). We propose that at least some of the regulatory functions ascribed to these proteins are performed by the Nup98-DHX9 complex. Several results support this conclusion including our findings that both proteins associate with similar gene loci and their RNA products. For example, our examination of previously published data sets (*Chen et al., 2014*; *Erkizan et al., 2015*; *Franks et al., 2016*; *G Hendrickson et al., 2016*) revealed a strong correlation between gene loci bound to Nup98 and the association of Nup98 and DHX9 with the mRNA products of these genes. Specifically, of the gene loci that interact with nucleoplasmic Nup98, 70% produce transcripts that are also bound to Nup98 ($p=6.14\times10^{-215}$) and ~27% produce transcripts bound to both Nup98 and DHX9 ($p=3.35\times10^{-58}$). Furthermore, our analysis of several putative DHX9-interacting gene loci revealed that Nup98 and DHX9 at these gene loci have related properties, including (1) Nup98 and DHX9 require one another for binding to these genes (*Figure 10*; *Figure 10—figure supplement 1*), (2) the loss of either protein alters the normal expression of these genes (*Figure 11*), and (3) the binding of Nup98 and DHX9 to the transcripts of these genes is interdependent on one another (*Figure 9*).

Of note, many of the genes showing altered expression following depletion of Nup98 or DHX9 contain a putative cAMP-response element (CRE) (see Results). CRE regulated genes represent ~50% of the Nup98 interacting gene loci detected in Nup98-Dam-ID studies (*Franks et al., 2016*), and of these genes ~72% have their transcripts bound by Nup98 ($p=4.2\times10^{-205}$) and ~36% bound by DHX9 ($p=2.3\times10^{-5}$). Consistent with these observations, both Nup98 and DHX9 have been reported to bind to the CREB-binding protein (CBP)/p300 (*Aratani et al., 2001*; *Kasper et al., 1999*; *Nakajima et al., 1997*), a transcriptional co-activator (*Vo and Goodman, 2001*).

On the basis of their physical and functional links to CRE regulated genes, we used a CRE-luciferase reporter assay to assess the role of Nup98 in DHX9-mediated transcription. Aratani and colleagues previously used this assay to characterize two modes of DHX9-stimulated reporter expression, one dependent on, and the other independent of, DHX9 ATPase activity (*Aratani et al., 2001*). Using DHX9 point mutants that either reduce or eliminate its ATPase activity (*Aratani et al., 2001*), we assessed the ability of Nup98 to stimulate the ATPase activity of DHX9 and modulate its transcriptional activity. Importantly, the expression of Nup98, while itself unable to stimulate reporter expression, could suppress the transcriptional defects of the DHX9 point mutant with reduced ATPase activity (DHX9[I347A]), but not an ATPase dead mutant (*Figure 12*). These results are consistent with our in vitro analysis showing Nup98 can stimulate the ATPase activity of DHX9 and supports the hypothesis that Nup98 functions as a cofactor to regulate the ATPase-dependent transcriptional functions of DHX9.

Nup98 is also predicted to stimulate the cellular activities of DHX9 that facilitate efficient processing and release of mRNAs from DHX9 (*Jankowsky, 2011*; *Jarmoskaite and Russell, 2014*). The latter idea is supported by our data showing that depletion of Nup98 causes an increase in the binding of RNA to DHX9 (*Figure 9*). It is worth noting that the Nup98-stimulated release of DHX9 from transcripts may be required to maintain the stable association of DHX9 with a gene locus by preventing it from leaving with the mRNA transcript. This idea is consist with our data showing that in the absence of Nup98 the occupancy of DHX9 at genes regulated by these proteins is reduced while DHX9 association with the corresponding mRNAs is increased (*Figures 9* and *10* and *Figure 10— figure supplement 1*).

Given the RNA binding properties of the Nup98-DHX9 complex, and that transcription and mRNA splicing are often coupled, we envisage that defects associated with disruption of this complex would alter splicing. This proved to be the case as the analysis of cells depleted of Nup98 or DHX9 revealed shared splicing defects at the level of E1A reporter (*Figure 13*) and throughout the transcriptome (see Results). Our observations are the first to suggest a role for Nup98 in mRNA splicing, and are consistent with a previously proposed role for DHX9 in splicing regulation in mammals (*Hartmuth et al., 2002*) and Drosophila (*Pellizzoni et al., 2001a*; *Reenan et al., 2000*).

### Nup98 and RNA helicases beyond DHX9

Finally, we must make note that the role for Nup98 in regulating DHX9 may extend to other DExH/D-box proteins, since five other RNA helicases were identified as Nup98 interactors in our study, including DDX21, DDX5, DDX17, DHX15, and DDX3. DDX5 and DDX17 are highly similar proteins that act as corepressors and coactivators through their interaction and modulation of transcription factors (*Fuller-Pace, 2013*). Like Nup98 and DHX9, DDX21, DDX3, and DHX15 have also been implicated in the antiviral immune response by sensing viral dsRNA and contributing to the regulation of expression of interferon and interferon-stimulated genes (*Fullam and Schröder, 2013*; *Lu et al., 2014*; *Wang et al., 2015*). Consequently, it will be important to determine whether Nup98 modulates the activity of these other RNA helicases and, in this context, plays a more general role in regulating RNA processing.

## Materials and methods

### Plasmids

Plasmids used in this study are described in *supplemental file 1B*, including DNA inserts, cloning sites, plasmid backbones and tags present.

### Antibodies

Nup98-specific antibodies (*Mitchell et al., 2010*) and anti-GFP rabbit polyclonal antibodies (*Makhnevych et al., 2003*) were previously described. Commercial antibodies include mAB414 against Nup62, Nup153, Nup214, and Nup358 (Abcam ab24609, RRID:AB_448181), α-tubulin (Sigma-Aldrich T6074, RRID:AB_477582), DHX9 (Abcam ab54593, RRID:AB_943711), hnRNP U (Abcam ab10297, RRID:AB_297037), PRKDC (Thermo Scientific MS-423-P1, RRID:AB_61152), GFP (Sigma-Aldrich 11814460001, RRID:AB_390913) and GST (GE Healthcare Life Sciences 27-4577-01, RRID:AB_771432). Goat anti–rabbit IgG-HRP (Bio-Rad 170–6515, RRID:AB_11125142), goat anti–mouse IgG-HRP (Bio-Rad 170–6516, RRID:AB_11125547), Alexa Fluor 750 goat anti-rabbit IgG (ThermoFisher Scientific A21039, RRID:AB_10375716), and Alexa Fluor 680 goat anti-mouse IgG (ThermoFisher Scientific A21057, RRID:AB_2535723) were used for Western blotting. Alexa Fluor 488 donkey anti-rabbit (ThermoFisher Scientific A21206, RRID:AB_2535792), Alexa Fluor 488 donkey anti-mouse (ThermoFisher Scientific A21202, RRID:AB_2535788), Alexa Fluor 594 donkey anti-mouse (ThermoFisher Scientific A21203, RRID:AB_2535789), and Alexa Fluor 594 goat anti-rabbit (ThermoFisher Scientific A11012, RRID:AB_10562717) were used for immunofluorescence microscopy.

### Cell culture

Cell lines used in this study include ATCC authenticated human HEK293T cells (HEK293T/17, ATCC: CRL-11268, RRID: CVCL_1926) and human U937 cells (ATCC: CRL-1593.2, RRID: CVCL_0007) (*Sundström and Nilsson, 1976*). HEK293T cells were grown in DMEM media (GIBCO 11965–092)

supplemented with 10% fetal bovine serum (GIBCO 12483–020) at 25% to 80% of confluency. U937 cells were cultured in RPMI 1640 media (GIBCO 22400–105) supplemented with 10% fetal bovine serum (GIBCO 12483–020) and kept at a cell density of $10^5$ to $2 \times 10^6$ cells/ml. Cell were determined free of mycoplasma contamination, as previously described (*Young et al., 2010*).

## Transfections

HEK293T cells were seeded in 100 mm tissue culture treated dishes ($3 \times 10^6$ cells/plate) 16 hours (hr) before transfection. Transfection of plasmid DNA was performed with TransIT LT1 (Mirus Bio LCC MIR 2300) transfection reagents per the manufacturer's protocol at 80% confluency. Twenty-four hours after transfection cells were collected for immunoprecipitation. The same transfection protocol described above was also performed in 24 well plates containing glass coverslips (Fisher Scientific, 12-545-80) and $5 \times 10^4$ cells/well, 24 hr before cells were prepared for immunofluorescence (described below).

## Immunoprecipitation procedures

Protein G Dynabeads (ThermoFisher Scientific 10004D) were conjugated to antibodies according to the manufacturer's instruction. Briefly, 200 µl of beads were conjugated to 10 µg of commercial anti-GFP, anti-DHX9, anti-hnRNP U mouse monoclonal antibodies (see previous section) or 10 µg of anti-Nup98 (*Mitchell et al., 2010*) or anti-GFP (*Makhnevych et al., 2003*). The mixture of beads and antibodies were prepared in 0.8 ml of PBS containing 0.02% Tween-20 (Sigma-Aldrich P9416-50ML) and incubated at room temperature for 10 minutes (min) with rotation. Beads conjugated to antibodies were resuspended with 200 µl of PBS with 0.02% Tween-20.

HEK293T cells or HEK293T cells transfected with *GFP*, *GFP-NUP98$^{1–920}$*, *GFP-NUP98$^{1–497}$*, or *GFP-NUP98$^{498-920}$* (see *Supplementary file 1B*) were detached from plates with trypsin (GIBCO, 25300–062) and washed twice with PBS (137 mM NaCl, 2.7 mM KCl, 4.3 mM $Na_2HPO_4$, 1.4 mM $KH_2PO_4$, pH 7.4). Cells were lysed with NP-40 cell lysis buffer (50 mM Tris, pH 7.4, 250 mM NaCl, 5 mM EDTA, 50 mM NaF, 1 mM $Na_3VO_4$, 1% Igepal CA-630, 0.02% $NaN_3$, and protease inhibitor cocktail [Sigma-Aldrich 11873580001]) on ice for 30 min using 400 µl of buffer per 100 mm diameter plate (approximately $6 \times 10^6$ cells/plate). Lysates were further disrupted by centrifugation through a QIAshredder column (QIAgen 79654) and samples were clarified by centrifugation at 14,000 X g for 20 min at 4°C. Cell lysate supernatant fractions were combined with antibody conjugated beads (50 µl of bead solution per 0.5 ml of cell lysate supernatant derived from $6 \times 10^6$ cells) and incubated at 4°C for 1 hr with rotation. Protein complexes bound to the beads were washed five times with 0.5 ml of NP-40 cell lysis buffer for 10 min at room temperature. Protein complexes were eluted from beads by heating to 100°C for 3 min in 25 µl Laemmli sample buffer with DTT (2% SDS, 10% glycerol, DTT, 0.01% bromophenol blue, 0.2 M of DTT and 0.06 M Tris-HCl, pH 6.8) per 50 µl of bead solution. Eluted proteins were analyzed by SDS-PAGE and western blotting as described below.

Immunoprecipitation reactions from HEK293T cells transfected with *GFP-NUP98$^{1–920}$* that were submitted to LC-MS/MS were prepared according to the protocol described above, but were scaled up ~4 fold (200 µl of bead solution per 2 ml of cell lysate supernatant derived from $3 \times 10^7$ cells).

Immunoprecipitation of Nup98 from U937 cells that were submitted to LC-MS/MS followed the same protocol described above with the following modifications. U937 cells were grown in suspension to $2 \times 10^6$ cells/ml in 100 mm culture plates. Harvested cells ($1 \times 10^8$) were lysed in 4 ml of NP-40 cell lysis buffer and samples were clarified by centrifugation. Cell lysate supernatant fractions derived from $5 \times 10^7$ cells (2 ml) were combined with 200 µl of antibody conjugated beads and incubated at 4°C for 1 hr. Bound protein complexes were washed, eluted and processed for SDS-PAGE as described above.

When performing RNase treatment of immunoprecipitated complexes, HEK293T cells ($1.2 \times 10^7$ cells) were lysed in 800 µl of NP-40 cell lysis buffer containing 40 µl of RNase OUT RNase Inhibitor (ThermoFisher Scientific, 10777–019) on ice for 30 min. Lysates were further disrupted by centrifugation through a QIAshredder column, and samples were clarified by centrifugation at 14,000 X g for 20 min at 4°C. Cell lysate supernatant fractions were combined with antibody conjugated beads (100 µl of bead solution and 800 µl of cell lysate supernatant) and incubated at 4°C for 1 hr. Protein complexes bound to the beads were washed three times with 1 ml of NP-40 cell lysis buffer without RNase OUT RNase Inhibitor. Samples were then split and half of the beads were resuspended in

NP-40 cell lysis buffer with RNase OUT RNase Inhibitor (100 µl) and the other half was incubated with 100 µl of NP-40 cell lysis buffer supplemented with 1 µl of the RNase A (10 mg/ml) (Thermo Fisher Scientific EN0531). Both samples were incubated at 37°C for 30 min and then washed three times with 0.5 ml of NP-40 cell lysis buffer for 10 min at room temperature. Protein complexes were eluted from the beads by the addition of 25 µl Laemmli sample buffer with DTT and heating at 100°C for 3 min.

## Immunoprecipitation from HeLa cell nuclear envelope and nucleoplasm fractions

Nuclei from HeLa cells (Sigma-Aldrich ECACC 93021013, RRID:CVCL_0030), were kindly provided by Dr. Paul Melançon (University of Alberta), and isolated from $10^7$ cells according to a previously published protocol (*Balch et al., 1984*). Pelleted nuclei were resuspended by drop-wise addition of 250 µl of ice-cold buffer A (0.1 mM $MgCl_2$, protease inhibitor cocktail) and vortexing. Nuclei were then immediately diluted by the addition of 1 ml of ice-cold buffer B (10% sucrose, 20 mM triethanolamine (pH 8.5), 0.1 mM $MgCl_2$, 1 mM DTT, and protease inhibitor cocktail) containing 1 µg/ml DNAse I (Sigma-Aldrich D5025) and incubated at room temperature for 15 min. A 10 µl sample of total nuclei was removed for western blotting and diluted with 10 µl of PBS. The nuclei suspension was centrifuged at 4100 X *g* for 15 min at 4°C to separate the nuclear envelope (pellet) and nucleoplasm (supernatant) fractions. A 20 µl sample of the nucleoplasmic fraction was removed for western blot and the remaining used for immunoprecipitation. The nuclear envelope (pellet) was resuspended in 1.24 ml of ice-cold buffer C (10% sucrose, 20 mM triethanolamine, pH 7.5, 0.1 mM $MgCl_2$, 1 mM DTT and protease inhibitor cocktail). A 20 µl sample was taken for western blot and the remainder used for immunoprecipitation.

One tenth volume of NP-40 cell lysis buffer stock solution (250 mM Tris-HCl, pH 7.5, 1.25 M NaCl, 25 mM EDTA, 5% NP-40, 5 mM VRC and protease inhibitor cocktail) was added to the nuclear envelope and nucleoplasm fractions and samples were divided into three equal volume samples. Each sample received 10 µg of anti-Nup98, anti-DHX9, or anti-GFP antibody. Anti-GFP antibodies were used in negative control immunoprecipitation reactions and are identified as control IgG in figures. Samples were incubated at 4°C with rotation for 1 hr. 100 µl of Protein G Dynabeads was then added and samples were incubated with rotation at 4°C for an addition 30 min. Beads were washed five times with 400 µl of NP-40 cell lysis buffer (25 mM Tris-HCl pH 7.5, 125 mM NaCl, 2.5 mM EDTA, 0.5% NP-40, 0.5 mM VRC and protease inhibitor cocktail). Samples were eluted into 40 µl of SDS-PAGE sample buffer by heating to 100°C for 3 min and analyzed by western blotting.

## SDS-PAGE and western blotting

Proteins resolved by SDS-PAGE were either stained with BioSafe Coomassie Stain (Bio-Rad 161–0786) or silver nitrate (Sigma-Aldrich S6506) to detect proteins or transferred to nitrocellulose membranes (0.2 µm, Bio-Rad 9004-70-0) for western blotting. These membranes were blocked in 5% skim milk in PBS-T (PBS containing 0.1% Tween 20) and incubated overnight at 4°C with the appropriate primary antibodies. Secondary antibodies conjugated to HRP or fluorescent dyes were used to visualize primary antibody binding.

## Mass spectrometry

Proteins present in immunoprecipitates of GFP-Nup98$^{1–920}$ from HEK293T cell lysates and endogenous Nup98 from U937 cell lysates were used for mass spectrometry analysis, as previously described (*Mitchell et al., 2010*). Briefly, proteins were resolved by SDS-PAGE, stained with Bio-Safe Coomassie Stain (Bio-Rad 1610786), bands excised from gel lanes, and subjected to in-gel trypsin digestion followed by LC-MS/MS using a mass spectrometer (Q-TOF Premier; Waters Corp.). Protein identification was performed by peptide mass fingerprinting using PEAKS mass spectrometry software (Bioinformatics Solutions, Inc.).

## Creating and analyzing protein-protein interaction networks

Curated protein-protein interactions (PPI) among identified Nup98 binding partners were extracted using the Search Tool for the Retrieval of Interacting Genes/Proteins (RRID:SCR_005223) (*Szklarczyk et al., 2011*). Only PPIs from curated databases or curated published experiments were

included in the PPI retrieval, and a minimum integrated confidence score of 0.5 was required for each interaction (for details see [*von Mering et al., 2003*]). Identified interactions were visualized using Cytoscape (RRID:SCR_003032) (*Smoot et al., 2011*). The PPI network edge thickness (*Figure 2*) represents the integrated confidence score for the interaction (ranging for medium confidence score of 0.5 to high confidence score of 1). Node colour in gray scale from light to dark indicates increasing abundance of the interactor in the GFP-Nup98 immunoprecipitation, based on number of unique peptides present in LC-MS/MS data (ranging from 5 to 30 unique peptides). Clustering of the resulting PPI network with the Cytoscape plugin MCODE (*Bader and Hogue, 2003*) identified highly interconnected proteins, which are likely to represent protein complexes in PPI network. We used the BinGO Cytoscape plugin (RRID:SCR_005736) (*Bader and Hogue, 2003*; *Maere et al., 2005*) to perform GO annotation enrichment analysis on the protein clusters identified by the MCODE plugin to infer biological processes for protein complexes. The node clusters identified by MCODE, representing putative protein complexes, along with the biological processes identified as enriched for each complex are indicated on the network as a coloured Venn diagram.

Network and node level statistics were extracted from the resulting network using Cytoscape. Specifically, network level statistics refers to statistically significant (p-value < 0.001) protein-protein interaction enrichment, comparing the number of interactions observed in the network to the expected number of interactions from a random graph with the same number of nodes. Node level statistics refers to node degree, thats is, the number of edges connected to each node in the network. Node degree was used as a selection criterion for which identified Nup98 interactors would be further investigated (see *Figure 2B*).

## Production of lentiviral pseudoparticles and lentivirus-induced protein depletion

Lentiviral pseudoparticles were produced in HEK293T cells ($2.5 \times 10^6$ cells) in tissue culture treated plates (100 mm diameter) as previously described (*Neufeldt et al., 2013*; *Schoggins et al., 2011*). The sequence of the shRNAs encoded in the pLKO.1puro plasmids (Sigma-Aldrich Mission shRNAs SHCLNG-NM_005387, SHCLNG-NM_001357, SHCLNG-NM_031844 or SHC002) are shown in the *Supplementary file 1C*. Samples indicated as control contain an shRNA sequence targeting a non-mammalian transcript. For lentivirus-induced protein depletion, cells were incubated with viral particles for 24 hr before the media containing lentiviral pseudoparticles was replaced by DMEM with 10% FBS. Cells were cultured for approximately 60 hr after transduction and then seeded into 24 well plates ($5 \times 10^4$ cells/well), wells containing cells for immunofluorescence contained glass coverslips. Ninety-six hours after transduction cells were prepared for immunofluorescence or collected in 200 μl of Laemmli sample buffer with DTT for SDS-PAGE and western blotting.

## Immunofluorescence

Glass coverslips containing HEK293T cells (described above) were washed twice with PBS, fixed for 10 min at room temperature with 3.6% formaldehyde (Sigma-Aldrich F8775) in PBS, washed twice with PBS, and then permeabilized for 2 min at room temperature with PBS containing 0.2% Triton X-100 (ThermoFisher Scientific BP151-500). Following two washes with PBS, samples were blocked in 2.5% skim milk in PBS-T for 2 hr at 4°C, probed with primary antibodies diluted in 2.5% skim milk in PBS-T overnight at 4°C, washed 3 times for 10 min with PBS-T, probed with secondary antibodies diluted in 2.5% skim milk in PBS-T for 2 hr at 4°C, and then washed 3 times for 10 min in PBS-T before mounting onto microscope slides (Fisher Scientific, 12-550-15) using DAPI-Fluoromount-G (Southern BioTech 0100–20). Epifluorescence images were acquired with an Axio Observer Z1 microscope, 63x/1.40 NA Oil UPlanS-Apochromat objective lens (Carl Zeiss, Inc.) and analyzed using Axiovision software (Carl Zeiss, Inc.) and Image J (*Schneider et al., 2012*).

## Recombinant protein expression and purification

Expression and purification of recombinant proteins was performed as previously described (*Mitchell et al., 2010*). Briefly, *E. coli* BL21-CodonPlus(DE3) cells (Agilent Technologies 230245) were transformed with the pGEX-6P-1 based plasmids (*Supplementary file 1B*), grown to an O. D.$_{600}$ of 0.6, and protein expression induced with 1 mM IPTG (ThermoFisher Scientific, BP175510) for 2 hr at 37°C (Nup98) or overnight at 16°C (DHX9). After collection by centrifugation, bacterial

cells were resuspended in lysis buffer (50 mM Tris, pH 7.5, 300 mM NaCl, 150 mM KOAc, 2 mM MgOAc, 10% glycerol, 0.1% Igepal CA-630, 1 mM DTT, and protease inhibitor cocktail) and sonicated. The soluble fractions of the lysates were cleared by centrifugation at 27,000 X g for 20 min. Purification of recombinant GST fusion proteins using glutathione–Sepharose 4B Media (GE Healthcare Life Sciences 17-0756-01) was performed as previously described (*Mitchell et al., 2010*). When appropriate, the GST tag was cleaved from the recombinant proteins using PreScission Protease as described by the manufacturer (GE Healthcare Life Sciences 27-0843-01).

## In vitro binding assays

Protein G Dynabeads (300 µl) were conjugated to 30 µg of anti-DHX9 antibody as described by the manufacturer and incubated with approximately 3.6 nmoles of GST-tagged DHX9 in 1.2 ml of PBS-T at room temperature for 1 hr with rotation. After washing to remove unbound protein, beads were resuspended in a total volume of 1.2 ml of PBS-T and 400 µl aliquots were incubated for 10 min at room temperature with either RNase A (final concentration 100 µg/ml), poly I:C RNA (Sigma-Aldrich P1530) (final concentration 100 µg/ml), or buffer alone. In parallel, GST-tagged Nup98 (1.2 nmoles in 200 µl of PBS-T) and purified GST (1.2 nmoles in 200 µl of PBS-T) were similarly treated with RNase A, poly I:C RNA, or buffer alone. Each of the three bead bound samples of GST-tagged DHX9 were then divided into two equal parts and combined with similarly treated GST-tagged Nup98 (0.6 nmoles in 200 µl of PBS-T per sample) or GST alone (6 nmoles in 200 µl of PBS-T per sample) and incubated at 4°C with rotation for 30 min. The protein complexes were washed five times with 500 µl of PBS-T, eluted from beads with 15 µl of Laemmli sample buffer with DTT, and analyzed by SDS-PAGE and western blotting. The same procedure described above was also performed with DHX9 and Nup98 after GST tag removal by cleavage with PreScission Protease.

## Bead-halo assay

The bead-halo assay was performed as previously described (*Patel and Rexach, 2008*; *Zhou et al., 2013*), with some modifications. To prepare bait samples, 15 µl of protein G Dynabeads were conjugated to 1.5 µg of anti-DHX9 antibody, divided into two equal samples and approximately 150 pmoles of purified recombinant DHX9 in 15 µl of PBS-T or buffer alone was added to each. These samples were further divided into three equal parts for the addition of RNase A (1 µg), poly I:C RNA (1 µg), or PBS-T alone and all six bait samples were then incubated at 4°C for 30 min. Prey samples were prepared by mixing 300 pmoles of purified recombinant Nup98 with 2 µg of anti-Nup98 and 2 µg fluorescently tagged Alexa Fluor 488 donkey anti-rabbit antibody in a final volume of 45 µl in PBS-T. This sample was incubated for 10 min at room temperature and then divided into three equal parts (15 µl per sample) for addition of RNase A (2 µg), poly I:C RNA (2 µg) or PBS-T alone, followed by incubation at 4°C for 30 min. Bait and prey samples were mixed per their additives (RNase A, poly I:C RNA or buffer alone) and incubated together for 10 min at 4°C. To define the domains of DHX9 that mediate Nup98 binding, GST or GST-Nup98[1–863] was immobilized on Glutathione High Capacity Magnetic Agarose beads (Sigma-Aldrich G0924) as the bait. DHX9 domains (1–380, 381–820, and 821–1270) tagged with eGFP at the C-terminus of DHX9 acted as prey, with both bait and prey samples being treated and combined for binding as above. All bead samples were washed three times with 60 µl of PBS-T before acquisition of epifluorescence images as described for immunofluorescence.

Image analysis was performed using ImageJ (RRID:SCR_003070) (*Schneider et al., 2012*) with custom macros (available upon request). Data processing was done in R (RRID:SCR_001905) (*R Core Team, 2016*). Briefly, images were opened in ImageJ and processed for background subtraction. Masks were created to identify beads and fluorescence intensities were measured for each masked bead. Bead fluorescence intensity measurement files were imported into R and aggregate averages were calculated for different experimental conditions and biological replicates. The fluorescence intensity of negative control samples was subtracted from corresponding experimental conditions. Mean and standard deviation of arbitrary fluoresce units of biological replicates were calculated and plotted in bar graphs.

## DHX9 ATPase assay

ATPase reactions were carried out in 96 or 384 well plates at 37°C using a previously described enzyme-coupled assay (*Panaretou et al., 1998*). Each 50 or 100 µL reaction contained 25 mM HEPES, 1 mM phosphoenolpyruvate (Sigma-Aldrich P7127), 3 mM MgCl$_2$, 1 mM DTT, 2.5 µL of Pyruvate Kinase/Lactic Dehydrogenase enzymes (Sigma-Aldrich P0294), 0.5 mM NADH (Sigma-Aldrich N8129), 2 mM ATP (Sigma-Aldrich L510327) and 30–40 nM of purified recombinant DHX9. Where specified, the reaction also contained 100 µg/ml poly I:C RNA (indicated as RNA on figures) and/or purified recombinant Nup98, GST-Nup98$^{1-920}$, GST-Nup98$^{1-497}$, GST-Nup98$^{498-920}$, or GST at an amount equimolar to DHX9 or as indicated on the figure. Control reactions for each condition contained the same reagents and recombinant proteins except for DHX9. ATP hydrolysis was monitored indirectly using absorbance of NADH at 340 nm, which was measured each minute for 120 min using a BioTek Synergy four microplate reader. The decrease of NADH absorbance over time was subsequently converted to micromoles of ATP consumed as previously described (*Montpetit et al., 2012*). The specific activity of DHX9 was calculated by subtracting the ATP consumption rate of control reactions (i.e. no DHX9) and normalizing the corrected rate to the concentration of DHX9 present, resulting in the ATP hydrolysis rate of DHX9 per second.

## RNA immunoprecipitation

The RNA immunoprecipitation protocol described below is based on previously described assays (*Conrad, 2008*; *Jensen and Darnell, 2008*; *Licatalosi et al., 2008*). Briefly, HEK293T cells were seeded in 150 mm diameter tissue culture plates, grown for 48 hr to ~75% of confluency (~10$^7$ cells), washed once with PBS, and cross-linked with 0.5% formaldehyde in PBS for 10 min at room temperature under slow shaking (approximately 70 rpm). Cross-linking was quenched with 220 mM glycine pH 7.0 for 5 min at room temperature, and cells were harvested by scraping and centrifugation (700 X g for 3 min at 4°C). Cells were washed four times with ice-cold PBS before lysis in 500 µl of lysis buffer (1.06 mM KH$_2$PO$_4$, 155 mM NaCl, 2.97 mM Na$_2$HPO$_4$, 0.1% SDS, 0.5% sodium deoxycholate, 0.5% Igepal CA-630, protease inhibitor cocktail and two units/µL of RNaseOUT Recombinant Ribonuclease Inhibitor). Cell lysates were spun through a QIAshredder spin column twice and the insoluble fraction was cleared by centrifugation at 16,000 X g for 10 min at 4°C. Samples of the input (10%) were removed for quantitation of total RNA and proteins present in the cleared cell lysates, and the remaining 90% of the samples were used for immunoprecipitation. For each immunoprecipitation reaction, 100 µl of Protein G Dynabeads was conjugated to 5 µg of anti-DHX9, anti-Nup98, or anti-GFP antibodies. Anti-GFP rabbit polyclonal antibodies were used as a negative control in immunoprecipitation reactions and are identified as control IgG in figures. Cleared cell lysates were added to antibody conjugated beads and incubated with rotation at 4°C for 2 hr. Beads were washed at 4°C with 1 ml of PBS cell lysis buffer, twice with 1 ml of high-salt buffer (5.3 mM KH$_2$PO$_4$, 775 mM NaCl, 14.18 mM Na$_2$HPO$_4$, 0.1% SDS, 0.5% sodium deoxycholate, 0.5% Igepal CA-630, protease inhibitor cocktail, and two units/µL of RNaseOUT Recombinant Ribonuclease Inhibitor), and then three times with 1 ml of PBS cell lysis buffer. Sample cross-linking was reversed by incubating beads with 140 µl of reverse buffer (10 mM Tris–HCl, pH 6.8, 5 mM EDTA, 10 mM DTT and two units/µL of RNaseOUT Recombinant Ribonuclease Inhibitor) for 45 min at 70°C. Input samples were also incubated with 120 µl of reverse buffer for 45 min at 70°C. Ten percent of each sample was removed for SDS-PAGE and western blotting. The remaining sample was treated with an equal volume of 2 x Proteinase K solution (0.2 mg/ml proteinase K, 40 mM Tris–HCl pH 7.5, 5 mM EDTA, 33.4 ng/ml Glyco-Blue (ThermoFisher Scientific AM9515), 0.2 mg/ml total yeast RNA (Sigma-Aldrich R6625)) for 30 min at 37°C to digest protein bound to the beads and release RNA. RNA was subsequently purified using TRIzol LS Reagent (ThermoFisher Scientific 10296–010) and treated with DNase I (ThermoFisher Scientific 18068015) before quantification. Reverse transcription reactions were performed on 1 µg of purified RNA using random primers (ThermoFisher Scientific 48190011) and Superscript II reverse transcriptase kit reagents (ThermoFisher Scientific 18064014) in a total volume of 20 µl. cDNA from the reverse transcription reaction (5 µl) was used as template in a 25 µl PCR reaction using primers described in the *Supplementary file 1D* and Phusion High-Fidelity DNA Polymerase (New England BioLabs, M0530S). PCR amplification products were resolved in 2% agarose gels in TBE buffer (100 mM Tris, 90 mM boric acid, and 1 mM EDTA) and visualized with SYBR Safe DNA

Gel Stain (ThermoFisher Scientific, S33102) in a Safe Imager 2.0 Blue Light Transilluminator (Thermo-Fisher Scientific, G6600).

## RNA immunoprecipitation from cells expressing shRNAs

RNA immunoprecipitations were performed as described in the previous section, with the following additions. HEK293T cells were seeded into 100 mm diameter tissue culture plates ($3 \times 10^6$ cells/plate) and transduced with lentiviral pseudoparticles encoding shRNAs targeting Nup98, DHX9, or non-mammalian (control) mRNAs. Cells were cultured ~60 hr after transduction, seeded into 150 mm diameter plates ($5 \times 10^6$ cells/plate) and ninety-six hours after transduction cells were cross-linked, RNA immunoprecipitations performed, and cDNA made. Real-time PCR (qPCR) was performed with the resulting cDNA using SYBR green super mix (Quanta 95055–100), per the manufacturer's protocol, in a Mx3000P QPCR System (Agilent Technologies 401403). All qPCR primers (shown in *Supplementary file 1D*) were designed using Primer3Plus software (*Untergasser et al., 2007*). Real time PCR results were analyzed as described (*Hellemans et al., 2007*). Change in the level of each specific mRNA bound to DHX9 or Nup98 is represented as a fold-change, between depleted and mock-depleted cells, in the ratio of bound:total mRNA of each transcript examined (i.e. transcript amount present in IP sample / transcript amount present in input sample). Thus the fold changes in the ratios of bound:total amount of any given transcript account for changes in the levels of that transcript in the depleted cells.

## DamID assay in mammalian cell culture

DamID assays were performed as previously described (*Franks et al., 2016*; *van Steensel*; *Vogel et al., 2007*). Nup98, Nup98$^{1-504}$, and GFP cloned into MSCV-DamID-Gateway plasmid and the pCL-Ampho plasmid were kindly provided by Drs. Tobias Franks and Martin Hetzer (Salk Institute for Biological Studies, CA, USA) and have been described (*Franks et al., 2016*). To produce Dam-DHX9, the DHX9 ORF in a pShuttle vector (GeneCopoeia, GC-H1793) was recombined into the MSCV-DamID-Gateway plasmid using the Gateway LR Clonase II Enzyme Mix (ThermoFisher Scientific 11791020) per the manufacturer's protocol.

Retroviruses encoding the Dam constructs described above were generated by co-transfection of pCL-Ampho plasmid and the MSCV-DamID vector of choice (5 μg of each plasmid) into HEK293T cells, using Lipofectamine 3000 (Thermo Fisher Scientific L3000008) with media replacement 6 hr after transfection. Two days after transfection, medium containing retroviruses was collected from HEK293T cells. Retroviruses were then added to naive HEK293T cells in six well plates ($6 \times 10^5$ cells/well) and incubated for 6 hr in the presence of hexadimethrine bromide (Sigma-Aldrich). To select cell lines stably expressing the Dam constructs, two days after transduction cells were switched to medium containing 1.5 μg/ml puromycin (Thermo Fisher Scientific A1113803) for a minimum of 10 days.

Stable cell lines expressing Dam constructs were transduced with lentiviral pseudoparticles encoding shRNAs targeting Nup98, DHX9 or a control sequence as described above. Protein depletion was allowed to proceed for 6 days and depletion was verified by western blotting. DNA was harvested from $2 \times 10^6$ cells using the Qiagen DNAeasy blood and tissue kit as described by the manufacturer (QIAgen 69504). Purified DNA (2.5 μg) was digested in a 10 μl reaction containing 0.5 μl of DpnI restriction enzyme (New England Biolabs R0176S) and CutSmart Buffer overnight. DpnI-digested DNA was ligated with a DamID adapter primer duplex (*Supplementary file 1D*) in a 20 μl ligation reaction with T4 DNA ligase (New England Biolabs M0202S) for 4 hr at 16°C. The ligation reaction was digested with DpnII (New England Biolabs R0543S) in a 50 μl reaction for 1 hr. Ten microliters of the DpnII digested ligation sample was amplified by PCR with Expand High Fidelity PCR System as described by the manufacturer (Sigma-Aldrich 11732641001). The resulting amplified DNA was purified with QIAquick PCR Purification Kit (QIAgen 28104) and used as template for real-time PCRs, as described above. Data from DamID qPCR was normalized to background amplification of genomic gene desert regions (*Supplementary file 1D*) and is shown as Dam-Nup98, Dam-Nup98$^{1-504}$ or Dam-DHX9 enrichment over Dam-GFP control.

## Quantification of nuclear and cytoplasmic transcripts

HEK293T cells were transduced with lentiviral pseudoparticles encoding shRNAs targeting Nup98, DHX9 or a control sequence as described above. Four days after transduction, $5 \times 10^6$ cells were collected and processed using the PARIS Kit (Thermo Fisher Scientific AM1921). The manufacturer's protocol for protein and RNA purification from cultured cells was used to produce protein and RNA samples from total cell extract and nuclear and cytoplasmic fractions. Fractionation efficiency was evaluated by western blotting. RNA samples were treated with DNase I and then reverse transcribed, before being used as template in real-time PCR reaction, as described above. Changes in the levels of each specific mRNA in the nuclear and cytoplasmic fraction are represented as a fold-change, between depleted and mock-depleted cells, in the ratio of the amount of an mRNA species present in the nuclear or cytoplasmic fraction to the total transcript amount present in cell lysates.

## Luciferase assay

HEK293T cells ($10^5$ cells per well in 24 well plates) were transfected with 455 ng of pGL4.29 [luc2P/CRE/Hygro] plasmid (firefly luciferase gene under control of a cAMP-response element (CRE); Promega E8471) and 45 ng of pGL4.75 [hRluc/CMV] plasmid (renilla luciferase gene under control of the CMV promoter) (Promega E6931). Cells were simultaneously co-transfected with 500 ng of pEGFP-C1 or pEGFP-NUP98$^{1–920}$ and 500 ng of pcDNA3 or pcDNA3-HA-DHX9 constructs (DHX9$^{WT}$, DHX9$^{I347A}$, DHX9$^{K417R}$). The pcDNA3-HA DHX9 constructs were a kind gift from Dr. Toshihiro Nakajima (Tokyo Medical University, Japan) (*Aratani et al., 2001*). Transfections were performed with Lipofectamine 3000. Samples were collected 24 hr after transfection in Passive Lysis Buffer as described by the manufacturer (Promega E1941). Luciferase activity was quantified in a BioTek Synergy four microplate reader using the Dual-Luciferase Reporter Assay System (Promega E1910) and data was analyzed as recommended by the manufacturer (*Schagat et al., 2007*). Briefly, the activity of firefly luciferase was normalized to the activity of renilla luciferase in the same sample. The normalized firefly luciferase activity of lysates from HEK293T cells, transfected with pGL4.29 and pGL4.75, were normalized to one and used to calculate the relative firefly luciferase activity for HEK293T cells co-transfected with pGL4.29, pGL4.75, and the various Nup98 and DHX9 constructs described above.

## Quantitation of transcript abundance and E1A mRNA splice isoforms

HEK293T cells were seeded into 24 well plates ($5 \times 10^4$ cells/well) 16 hr before transduction with lentiviral pseudoparticles. Cells were cultured approximately 60 hr after transduction, seeded into 12 well plates ($10^5$ cells/well), and ninety-six hours after transduction total RNA was purified from cells using Trizol Reagent (ThermoFisher Scientific 15596026). Purified RNA samples were treated with DNase I and reverse transcription and qPCR reactions were performed as described above. Real time PCR results were analyzed as described previously (*Hellemans et al., 2007*), with two reference genes (HPRT and GAPDH) for transcript abundance normalization and four reference genes (HPRT, GAPDH, ACTB, TUBA1A) used for normalization of splice isoforms. The relative quantity of each E1A splice isoform was normalized to the total E1A transcripts present in the same sample (determined using an E1A primer amplifying all splice isoforms and pre-mRNA). Fold changes in the abundance of different E1A splice isoforms upon Nup98 or DHX9 depletion was calculated relative to isoform abundance in cells transduced with control shRNA (mock depleted cells).

## Comparative analysis of large scale sequencing datasets

Large scale datasets were obtained from the National Center for Biotechnology Information (NCBI) Gene Expression Omnibus (GEO) repository (RRID:SCR_005012), unless otherwise stated.

Genome-wide Nup98 interaction with chromatin was assessed through available DamID-seq data by comparing enriched DNA sequences from Dam-Nup98 or Dam-Nup98$^{1–504}$ expressing HeLa-C cells to those of Dam-GFP expressing cells (GSE83692). Data analysis was performed as described in the corresponding dataset and its publication (*Franks et al., 2016*).

Transcriptome-wide interaction of Nup98 with mRNA was determined from available sequencing data for Nup98 RNA immunoprecipitations from K562 cells (GSE67963) (*G Hendrickson et al., 2016*). DHX9 interaction with RNA was determined from sequencing data for DHX9 RNA immunoprecipitation from TC32 cells, kindly provided by Drs. Hayriye Erkizan and Jeffrey Toretsky (Georgetown University) (*Erkizan et al., 2015*). Data analysis was performed as described in the

corresponding datasets and their indicated publication, transcripts were considered as interacting with target proteins if showing a fold enrichment above 1.5 and adjusted p-value < 0.05.

Transcriptome-wide changes in transcript or splicing isoform abundance were determined from RNA-sequencing data for HepG2 or IMR90 cells upon Nup98 depletion (GSE83551) (*Franks et al., 2016*). Transcriptome changes in NB1 cells upon DHX9 depletion were determined from available RNA-sequencing data (GSE44585) (*Chen et al., 2014*). Transcriptome sequencing data were analyzed as previously described (*Wolfien et al., 2016*), using Galaxy (RRID:SCR_006281) (*Afgan et al., 2016*), R (*R Core Team, 2016*), and Bioconductor (RRID:SCR_006442) (*Huber et al., 2015*). An adjusted p-value < 0.05 was used to identify transcripts/isoforms whose abundance was significantly altered upon target protein depletion. The transcripts/isoforms whose abundance was significantly altered upon Nup98 depletion in the two above mentioned datasets (HepG2 or IMR90 cells) were combined as a single dataset to facilitate comparisons with DHX9 depletion datasets.

All datasets were aligned to human reference sequence GRCh37/hg19 and annotated with corresponding UCSC genes (RRID:SCR_005780) and Ensembl genes/transcripts (RRID:SCR_002344) (*Huang et al., 2013*; *Rosenbloom et al., 2015*; *Yates et al., 2016*; *Yu et al., 2015*). Statistically significant overlap between gene sets were calculated using the Fisher's exact test based on the hypergeometric distribution through the R package GeneOverlap (*Shen, 2013*). More information on the GeneOverlap R package is available from the Bioconductor website (https://www.bioconductor.org/). More information on the analysis and data set comparisons described above, including R scripts, galaxy workflows and lists of genes present in each dataset, is available at *Capitanio, 2017* (with a copy archived at https://github.com/elifesciences-publications/Nup98_eLife).

# Acknowledgements

We thank those individuals mentioned in the in the Materials and methods section for providing reagents and data. We also thank Paul LaPointe (University of Alberta) and members of his laboratory for assistance with ATPase assays, and Anil Kumar (University of Alberta) for assistance with luciferase assays. We also thank past (Jana Mitchell and Christopher Neufeldt) and current members of the Wozniak and members of the Montpetit laboratory for reagents, thoughtful discussions, and experimental assistance. This work was supported by an Alberta Innovates Health Solutions graduate scholarship to JSC and Canadian Institutes of Health Research grants to BM (MOP 130231) and RWW (MOP 106502 and 36519).

# Additional information

## Funding

| Funder | Grant reference number | Author |
| --- | --- | --- |
| Alberta Innovates - Health Solutions | Graduate Studentship | Juliana S Capitanio |
| Canadian Institutes of Health Research | MRC Operating Grant Program,130231 | Ben Montpetit |
| Canadian Institutes of Health Research | MRC Operating Grant Program,106502 | Richard W Wozniak |
| Canadian Institutes of Health Research | MRC Operating Grant Program,36519 | Richard W Wozniak |

The funders had no role in study design, data collection and interpretation, or the decision to submit the work for publication.

## Author contributions

JSC, Conceptualization, Data curation, Formal analysis, Investigation, Visualization, Methodology, Writing—original draft, Writing—review and editing; BM, Conceptualization, Resources, Formal analysis, Supervision, Funding acquisition, Investigation, Visualization, Methodology, Project administration, Writing—review and editing; RWW, Conceptualization, Resources, Formal analysis, Supervision,

Funding acquisition, Investigation, Visualization, Methodology, Writing—original draft, Project administration, Writing—review and editing

## Author ORCIDs

Juliana S Capitanio, http://orcid.org/0000-0003-4283-8007
Ben Montpetit, http://orcid.org/0000-0002-8317-983X
Richard W Wozniak, http://orcid.org/0000-0003-2328-0247

## Additional files

### Supplementary files

• Supplementary file 1. Supplemental information. (A) Complete mass spectrometry data exported from PEAKS software. (B) List of plasmids. (C) List of shRNAs. (D) List of primers.

### Major datasets

The following previously published datasets were used:

| Author(s) | Year | Dataset title | Dataset URL | Database, license, and accessibility information |
|---|---|---|---|---|
| Franks T, Hetzer M | 2016 | DamID in HeLa-C cells | https://www.ncbi.nlm.nih.gov/geo/query/acc.cgi?acc=GSE83692 | Publicly available at the NCBI Gene Expression Omnibus (accession no: GSE83692). |
| Hendrickson DG, Kelley DR | 2016 | Widespread RNA Binding by Chromatin Associated Proteins | https://www.ncbi.nlm.nih.gov/geo/query/acc.cgi?acc=GSE67963 | Publicly available at the NCBI Gene Expression Omnibus (accession no: GSE67963). |
| Franks T, Hetzer M | 2016 | RNA-seq in HepG2 and IMR90 cells | https://www.ncbi.nlm.nih.gov/geo/query/acc.cgi?acc=GSE83551 | Publicly available at the NCBI Gene Expression Omnibus (accession no: GSE83551). |
| Chen ZX, Ramskold D, Schlisio S | 2014 | RNA helicase A is necessary for KIF1B$\beta$ tumor suppression in neuroblastoma | https://www.ncbi.nlm.nih.gov/geo/query/acc.cgi?acc=GSE44585 | Publicly available at the NCBI Gene Expression Omnibus (accession no: GSE44585). |
| Erkizan HV, Schneider JA, Sajwan K, Graham GT, Griffin B, Chasovskikh S, Youbi SE, Kallarakal A, Chruszcz M, Padmanabhan R, Casey JL, Üren A, Toretsky JA | 2015 | Supplementary Table 1 | https://www.ncbi.nlm.nih.gov/pmc/articles/PMC4333382/bin/supp_gku1328_Supplementary_Table1_Erkizan.xlsx | Publicly available at the NCBI PubMed Central (PMCID: PMC4333382). |

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
