## [Decision Letter]

Thank you for submitting your article "Human Nup98 regulates the localization and activity of DExH/D-box helicase DHX9" for consideration by *eLife*. Your article has been reviewed by three peer reviewers, one of whom is a member of our Board of Reviewing Editors, and the evaluation has been overseen by Kevin Struhl as the Senior Editor. The reviewers have opted to remain anonymous.

The reviewers have discussed the reviews with one another and the Reviewing Editor has drafted this decision to help you prepare a revised submission.

In this manuscript, Capitanio et al. use a proteomic approach to identify novel binding partners of Nup98. They identify multiple RNA binding and DEAD-box proteins including DHX9. They follow up on this observation and show that Nup98 and DHX9 can directly interact in vitro and that this interaction is enhanced by the presence of RNA. Furthermore, they show that Nup98 stimulates the ATPase activity of DHX9. Using shRNA depletion experiments they then attempt to characterize the functional significance of the NUP98-DHX9 interaction. Depletion of either NUP98 or DHX9 leads to changes in the splicing pattern of E1A mRNA, which interacts with both proteins.

After discussion, the three reviewers agreed that this manuscript contains potentially interesting observations. However, there were concerns about the functional importance of the interaction between Nup98 and DHX9 and several aspects of the manuscript need further development. The following key issues were identified that would have to be addressed before the paper could be suitable for *eLife*.

1) Clarification of the role of RNA for the interaction between DHX9 and NUP98 and demonstration that endogenous proteins interact directly. This is particularly important since in IPs with antibodies to Nup98 and DHX9, the two proteins do not appear to co-IP each other (Figure 8—figure supplement 1 and Figure 9—figure supplement 1).

2) Functional nature of the DHX9-NUP98 interaction.

The authors show that knock-down of either DHX9 or NUP98 causes splicing defects of the E1A mRNA. In order to show that the lack of interaction between these two proteins is responsible for this phenotype, the authors need to identify specific point mutants that interfere with the interaction, introduce them into cells and analyze the splicing defects.

3) A novelty of the manuscript lies in the potential off-pore or intranuclear function of Nup98, presumably in association with active genes. Yet there is no direct evidence that the interaction between Nup98 and DHX9 happens in the nucleoplasm, or that their interaction occurs in association with active genes. The authors have to test whether Nup98 or DHX9 bind to the genes of affected mRNAs or that their interaction is occurring in the nucleoplasm.

Reviewer #1:

Overall this is a solid manuscript of high technical quality. However, the functional aspects of the manuscript are somewhat weak, and the significance of the NUP98-DHX9 interaction remains uncertain since each protein alone affects E1A mRNA splicing. Here it would be important to introduce specific mutants that interfere with the interaction between these proteins without altering their steady state levels.

Reviewer #2:

The manuscript by Capitanio et al., "Human Nup98 regulates localization and activity of DExH/D-box helicase DHX9" introduces a novel mechanism by which nuclear pore proteins and specifically Nup98 can contribute to gene expression. The major findings of the manuscript include an identified direct interaction between Nup98 and an RNA helicase DHX9 and specific ways in which Nup98 contributes to known activities of DHX9, such as the ATPase activity, RNA binding and splicing. Importantly, the study also provides evidence for previously suggested but incompletely characterized functions of Nup98 in mRNA binding and RNA splicing. This is particularly interesting since Nup98 and other Nups have been repeatedly shown to contribute to transcription, and the reported roles of Nup98 in RNA processing may be part of this gene regulatory pathway. Although novel and potentially of wide interest, several conclusions in the manuscript require further evidence, controls and clarification, as detailed below.

Major points:

1) One of the conclusions in the paper (last sentence of the Summary, for example) states that the detected interactions between Nup98 and DHX9, and between Nup98 and RNA occur away from NPCs. This is based partly on the fact that DHX9 localizes primarily to the nucleoplasm and the Nup98-GFP and DHX9 co-localize in nucleoplasmic GLFG bodies (subsection “Nup98 as a positive regulator of the DBP DHX9” paragraph four, as an example). But the authors do not provide direct evidence that any of these interactions are not occurring at the nuclear pores. The IPs, which led to identification of DHX9 and of other interacting partners of Nup98, are carried out from whole cell lysates, thus it is quite possible that these interactions may represent events that occur at nuclear pores and as part of mRNA export. Furthermore, although Nup98-GFP appears to primarily localize to the intranuclear GLFG bodies (Figure 5), Nup98 may still also localize to nuclear pores, just at lower levels than into GLFG bodies. It seems critical for the authors' conclusions to provide additional evidence that the Nup98-DHX9 interaction occurs in the nucleoplasmic pool and is not primarily occurring at the NPC. The authors can perhaps fractionate their whole cell lysates and test whether GFP-Nup98 can IP DHX9 (as shown in Figure 1 and Figure 6) from nuclear soluble fractions or chromatin-enriched fractions.

A similar concern applies to presented interactions between Nup98, DHX9 and specific mRNAs (Figure 8). Can the authors also test that these interactions with RNA occur in the nucleus vs. the cytoplasm, and preferentially in the nuclear soluble fractions? Another approach may be to block nuclear pore transport or mRNA export and test whether these interactions still occur to a similar level.

2) Alternatively, the interactions presented here may be primarily occurring within GLFG bodies. In vivo interactions between Nup98 and DHX9 (and other interacting partners) are shown only for Nup98-GFP fusion. Although GLFG bodies may be related to some endogenous compartment of the nucleus, it is unclear how they relate to the endogenous functions of Nup98 since normally Nup98 is not seen in such large accumulations (Figure 4, for instance). The IPs of endogenous proteins, using antibodies to Nup98 and DHX9, do not appear to co-IP each other (Figure 8—figure supplement 1 and Figure 9—figure supplement 1). Although the authors provide strong evidence that Nup98 and DHX9 interact directly, it is puzzling that endogenous proteins do not co-IP. Can co-IP between Nup98 and DHX9 be attempted under different conditions? Or can the authors at least comment on this discrepancy?

3) Figure 9 shows that interactions between specific mRNAs and Nup98 increase upon DHX9 depletion. Again, there seems to be a possibility that this is a consequence of altered mRNA export and accumulation of these mRNAs at NPC-bound Nup98. Furthermore, the decreased interaction between DHX9 and mRNAs upon depletion of Nup98 in 9B may be a consequence of decreased transcription of these mRNAs due to the function of Nup98 in gene activation. To help distinguish these possibilities, it is essential that the authors show total levels of these transcripts and preferably nascent transcripts in these conditions, in addition to the immunoprecipitated mRNA amounts shown in the figure. Furthermore, it would be helpful to determine if export of these mRNAs is disturbed upon Nup98 or DHX9 depletion, by single RNA FISH or by RT-PCR on nuclear vs. cytoplasmic fractions.

4) Similar concern applies to Figure 10. The demonstrated role of Nup98 in alternative splicing is very interesting, but since the authors are focusing on gene expression mechanisms, they need to also assess transcription levels of E1A and whether Nup98 and DHX9 affect generation of the E1A nascent transcript or if their role is restricted to splicing regulation. This would be particularly informative in light of the known roles of Nup98 in binding to transcribing genes, which the authors discuss in the manuscript (subsection “Nup98 regulated gene expression via transcription and mRNA processing”). In fact, demonstration of binding of Nup98 or DHX9 to the E1A gene by ChIP-qPCR would also be highly supportive of the conclusions of the paper. Although perhaps not required for this work, it could provide an alternative means to provide further evidence that these interactions are not taking place at the nuclear pores as part of mRNA export.

Reviewer #3:

This work in this manuscript investigates the nucleoporin protein Nup98 for roles beyond those linked to the nuclear pore. Nup98 is shown to co-immunoprecipitate with a large set of proteins, including DHX9 and other RNA helicase proteins. The rest of the work focuses mainly on the interaction with DHX9 and makes the case that the interaction of Nup98 and DHX9 is direct and functionally important. Although the topic and the work are quite interesting, in my opinion the work falls short of making a large advance because the functional nature of the interaction between Nup98 and DHX9 is not demonstrated clearly. This conclusion is made for the following reasons.

1) The interaction may be mediated or bridged by RNA. When the interaction with Nup98 is localized to the N- and C-terminal domains of DHX9, binding is eliminated upon RNase A treatment (Figure 6), suggesting an RNA linker between the two proteins. For the interaction with the intact DHX9, treatment with RNase reduced but did not eliminate the interaction (Figure 6), leading to the suggestion that the two proteins interact directly. However, it was not established in these experiments that the RNase treatment was complete, leaving open the possibility that a more extensive RNase treatment would have eliminated the interaction altogether by destroying a linker RNA. It is also possible that the interaction is relatively weak. In the in vitro binding experiments the protein concentrations appear to be rather high (based on my calculations using amounts and volumes from the Methods section), well into the micromolar range.

2) There is little information about potential specific functions of a Nup98-DHX9 complex. The two proteins bind an overlapping set of RNAs, but this does not indicate a function for the protein-protein complex. In the experiments probing splicing intermediates of E1A, depletion of each protein influences the ratios of splicing intermediates, but the effects are not the same. Thus, the experiment indicates that both proteins influence splicing of E1A, but not that the two proteins interact functionally to achieve these effects. This conclusion is accurately reached in the statement, "Thus, both Nup98 and DHX9 appear to play a role in the regulation of E1A mRNA splicing.". The finding that Nup98 increases the RNA-dependent ATPase activity of DHX9 is interesting, but there is no information on how this acceleration is achieved or on the functional implications. It is also interesting that depletion of each protein influences the mRNAs that are bound to the other protein in vivo, but this relationship is not straightforward. Depletion of Nup98 decreases binding of several RNAs to DHX9, consistent with a simple mechanism of cooperative binding by the two proteins, but depletion of DHX9 increases mRNA levels bound to Nup98, contrary to the simplest model. Thus, one is left wondering about the functional implications or significance of these results.

Without clearer evidence for the nature and the functional roles of the Nup98/DHX9 complex, it is my opinion that the manuscript is probably better suited for a more specialized journal.

[Editors' note: further revisions were requested prior to acceptance, as described below.]

Thank you for resubmitting your work entitled "Human Nup98 regulates the localization and activity of DExH/D-box helicase DHX9" for further consideration at *eLife*. Your revised article has been favorably evaluated by Kevin Struhl (Senior editor) and three reviewers, one of whom is a member of our Board of Reviewing Editors.

The manuscript has been improved and all three reviewers support publication. However, there are three very minor remaining issues that need to be addressed before acceptance, as outlined below:

1) Subsection “Nup98 stimulates the ATPase activity of DHX9”, the authors state that the basal ATPase rate of their DHX9 was comparable to previous publications. An attempt by reviewer 3 to calculate the rate in one of these citations (Zhang & Grosse), returned a value of ~15-30 s^-1^ (depending on the RNA used), quite different from the 1 s^-1^ in the current work. However, the other citation (Schütz et al) reported a rate constant very similar to the current work. The authors may want check this earlier work.

2) The citation to "Aratani, 2006" should be the 2001 paper.

3) Subsection “Nup98 and DHX9 interact with a shared subset of mRNAs and gene loci.”, "Depletion of Nup98 or DHX9 significantly reduced the interactions of its binding partner with the target gene." This result here, with the DamID assay, seems not to fit in neatly with the result that "depletion of Nup98 led to a significant increase in the amount of each of the six mRNAs bound to DHX9." This isn't really dealt with directly by the authors. Is their model that the proteins bind cooperatively to DNA but bind RNA in a more complicated way, as noted "…consistent with a model in which DHX9 promotes the association of Nup98 with specific mRNAs, and Nup98 facilitates the release of these mRNAs from DHX9."

Reviewer #1:

Many concerns were addressed by additional experiments. While I still see some issues the manuscript was strengthened by these additions.

Reviewer #2:

I believe the authors have addressed the majority of the issues in a satisfactory way. Specifically, they now provide evidence 1) for physical interaction between endogenous Nup98 and DHX9 in vivo, and within the nucleoplasm; 2) for chromatin binding of Nup98 and DHX9 to at least a group of candidate genes, using DamID-qPCR, and transcription and splicing of some of these target genes also appear to be affected by levels of Nup98 and DHX9; 3) for the functional role of this interaction in at least transcriptional output of target genes by analysis of several mutations, although this is done with a reporter gene. Overall, the manuscript has clearly improved in several ways and now provides a more convincing and clear demonstration for the functional relationship of Nup98-DXH9 complex in transcription and RNA processing. I recommend for publication.

Reviewer #3:

The authors have made a serious effort to address the concerns of the previous reviews and the manuscript is improved substantially. In terms of the major points:

1) The role of RNA in the interaction of DHX9 and Nup98. I am satisfied that they have taken reasonable steps to rule out the presence of residual RNA in the RNase treated samples. The amount of RNaseA they used, 100 µg/ml for 10 min, is very likely to be sufficient to remove essentially all of the RNA, and they state that it removed all detectable RNA. They also state that pre-treating DHX9 and Nup98 with high concentrations of RNA does not block interaction as monitored by co-IP, which would be expected if each protein simply bound RNA independently.

2) Do Nup98 and DHX9 interact away from the nuclear pore? I am convinced here, both by the fractionation experiments and by the mutant Nup98 that does not associate with the pore but still binds DHX9.

3) Is the interaction of Nup98 and DHX9 functionally important? I think this is where the response is less convincing, and the case remains circumstantial. I found the experiments using ATPase mutants of DHX9 unconvincing, because although the authors conclude that increased production of luciferase activity by Nup98 required functional ATPase activity of DHX9, if one looks at the heights of the bars in Figure 12, the extent of stimulation by Nup98 was really approximately the same for all versions of DHX9 and at least as large, in terms of the fold effect, for the empty vector. In other words this could be an effect of Nup98 that is completely independent of DHX9. However, there are several other pieces of data suggesting the co-localization of DHX9 and Nup98 within the genome, including in regions where they both function, and the two proteins are also shown independently to interact. I think this is a very strong circumstantial case for a functional interaction.

---

## [Author Response]

After discussion, the three reviewers agreed that this manuscript contains potentially interesting observations. However, there were concerns about the functional importance of the interaction between Nup98 and DHX9 and several aspects of the manuscript need further development. The following key issues were identified that would have to be addressed before the paper could be suitable for eLife.

1) Clarification of the role of RNA for the interaction between DHX9 and NUP98 and demonstration that endogenous proteins interact directly. This is particularly important since in IPs with antibodies to Nup98 and DHX9, the two proteins do not appear to co-IP each other (Figure 8—figure supplement 1 and Figure 9—figure supplement 1).

“The role of RNA for the interaction between DHX9 and NUP98”. This point appears to have been raised based on the concerns of reviewer 3 and their inquiry into the conditions we have used for RNase digestion in the evaluation of the contribution of RNA to the Nup98-DHX9 complex. For these experiments, we have used RNase concentrations (100 µg of RNase/ml) above that considered sufficient to degrade protein-bound RNA and used in other studies (Höck et al., 2007; Moore et al., 2014; Ule, Jensen, Mele, & Darnell, 2005; Zhang et al., 2008). In fact, we observed that treatment of immunoprecipitated Nup98 or DHX9 with the RNase concentrations used in our experiments reduced RNA levels to non-detectable levels as assayed by gel electrophoresis or spectrometry. Thus, it seems reasonable to conclude that the RNase levels we have used in these experiments are sufficient to eliminate RNA. This is now discussed in the Results section.

Importantly, in each of the experiments in which we have examined the interactions of Nup98 and DHX9, including immunoprecipitations and in vitro analyses of recombinant proteins, we have observed that treatment of the complexes with RNase or inclusion of RNase in binding reactions, reduced, but did not eliminate, the amount of Nup98-DHX9 complex detected (Figure 6). Furthermore, the inclusion of RNA in the in vitro binding reactions increased the amount of complex detected in these assays, but it was not required for their association (Figure 6 and Figure 6—figure supplement 1). Moreover, pre-incubating the proteins with excess RNA did not block their association, which would be expected if these proteins were interacting by binding to RNA independently. We conclude that RNA directly or indirectly (e.g. by inducing a conformational change) augments the interactions of Nup98 with DHX9.

“Demonstration that endogenous proteins interact directly”. Reviewer 1 raised the question of whether endogenous Nup98 and DHX9 interact based on results shown in Figure 8—figure supplement 1 and Figure 9—figure supplement 1. We have now included additional data to address the reviewer request and we have revised the Results section to clarify what was likely a misunderstanding in the interpretation of these supplemental figures. To summarize our data on the interactions of these proteins, in Figure 1 we showed that DHX9 is bound to GFP-Nup98 immunopurified from cell extracts. This result was confirmed by western blotting using anti-DHX9 antibodies and this blot was presented in the initial submission and another iteration of this experiment is now presented in Figure 5—figure supplement 3. In the revised manuscript, we now present the immunopurification of endogenous Nup98 and DHX9, and evidence for their association with one another (Figure 2—figure supplement 1, Figure 5, and Figure 6). These experiments were conducted using conditions that maintain protein-protein interactions. In contrast, the results shown in Figure 8—figure supplement 1 and Figure 9—figure supplement 1 are of immunoprecipitation experiments performed under conditions specifically designed to disrupt protein-protein interactions while allowing detection of cross-linked mRNA to the immunoprecipitated protein. These experiments were performed to examine the mRNA binding properties of Nup98 and DHX9 independent of one another. The conditions used in these experiments were described in the Material and Methods section, however, it was an oversight on our part not to clearly discuss the rational for, and results of, these experiments. This has now been corrected and we apologize to the reviewers for this confusion.

2) Functional nature of the DHX9-NUP98 interaction.

The authors show that knock-down of either DHX9 or NUP98 causes splicing defects of the E1A mRNA. In order to show that the lack of interaction between these two proteins is responsible for this phenotype, the authors need to identify specific point mutants that interfere with the interaction, introduce them into cells and analyze the splicing defects.

As suggested by the Editor and reviewers, we have worked to further examine the function of the Nup98-DHX9 interaction, but have taken a different approach from that suggested by reviewer 1. The suggestion that we generate point mutations that affect the interactions of Nup98 and DHX9 would require the production of a large number of mutations across the ~400 amino acid domains that mediate their association, for which we lack structural information. These many mutants would need to evaluated using in vivo assays to assess their expression, stability, and their association with their binding partner (by immunopurification). Importantly, even those mutants that appear to meet the required standards and disrupt the Nup98-DHX9 binding may also alter additional binding interactions and other cellular processes; making it necessary to show that any phenotype arising from a point mutant can be suppressed by the alterations in the structure or expression of the binding partner. As this extensive series of experiments represents a separate study in and of itself, we chose to focus on the analysis of existing DHX9 point mutations and assess the ability of Nup98 to suppress functional defects associated with mutations.

As DHX9 and Nup98 have been linked to transcription, we expanded our analysis of the functional relationships between these proteins to their role in transcription. Specifically, we tested the role of Nup98 in DHX9-mediated transcription using a CRE- luciferase reporter assay. This assay was previously used to evaluate the role of DHX9 in transcription, including defining the contributions of its ATPase activity (Aratani et al., 2006). Point mutants in DHX9 that reduce (DHX9I347A) or eliminate (DHX9K417R) ATPase activity show reduced stimulation of reporter expression. Since our in vitro assays showed that Nup98 could stimulate the ATPase activity of DHX9, we tested whether overexpression of Nup98 could stimulate the DHX9-mediated expression of CRE- luciferase. Importantly, the expression of Nup98, while itself unable to stimulate reporter expression, could suppress the transcriptional defects of the DHX9 point mutant with reduced ATPase activity (DHX9I347A), but not an ATPase dead mutant (Figure 12). These results are consistent with the ability of Nup98 to stimulate the ATPase activity of DHX9 and supports the hypothesis that Nup98 functions as a cofactor to regulate the ATPase-dependent transcriptional functions of DHX9. These data are now part of the manuscript.

In addition, we further extend the functional relationships between Nup98 and DHX9 in transcription and splicing by:

1) Demonstrating that several gene loci and their transcripts that interact with Nup98 and DHX9 have similar changes in transcript abundance upon Nup98 or DHX9 depletion (Figure 11).

2) Showing, through the analysis of recently published RNA-Seq data (Chen et al., 2014; Franks et al., 2016), that a shared set of genes show altered transcription upon loss of DHX9 or Nup98 (287 genes with altered expression upon DHX9 or Nup98 depletion, p = 3.2 X 10^-36^). In addition, Nup98 and DHX9 depleted cells reveal a significant overlap in gene products exhibiting altered splicing upon depletion of each protein. DHX9 depletion altered the splicing of 866 genes, of these 217 genes also showed altered splicing upon Nup98 depletion (p = 2.0 x 10^-43^). See Results section.

3) Finding that many of the genes showing altered expression following depletion of Nup98 or DHX9 contain a putative cAMP-response element (CRE), consistent with reports that both Nup98 and DHX9 bind to the CREB-binding protein (CBP)/p300 (Aratani et al., 2001; Kasper et al., 1999; Nakajima et al., 1997). CRE regulated genes represent ~50% of the Nup98-interacting gene loci detected in Nup98- DamID studies, and transcripts from ~72% of these genes were detected bound by Nup98 (p = 4.2 X 10^-205^) and ~36% bound by DHX9 (p = 2.3 X 10^-5^). See Results section.

4) Reporting a significant correlation between gene loci bound to Nup98 and the association of Nup98 and DHX9 with the mRNA products of these genes. Specifically, of the gene loci that interact with nucleoplasmic Nup98, 70% produce transcripts that are also bound to Nup98 (p = 6.1 X 10^-215^) and ~27% produce transcripts bound to both Nup98 and DHX9 (p = 3.35 X 10^-58^). See Discussion section.

3) A novelty of the manuscript lies in the potential off-pore or intranuclear function of Nup98, presumably in association with active genes. Yet there is no direct evidence that the interaction between Nup98 and DHX9 happens in the nucleoplasm, or that their interaction occurs in association with active genes. The authors have to test whether Nup98 or DHX9 bind to the genes of affected mRNAs or that their interaction is occurring in the nucleoplasm.

We have now assembled several pieces of data, both previously presented and new, supporting our conclusion that Nup98 interacts with DHX9 in the nucleoplasm. First, we showed in Figure 5 that production of GFP-Nup98 leads to its accumulation in multiple nucleoplasmic foci and DHX9 is recruited to these foci. Second, we showed that depletion of Nup98, while having no detectable effect on nuclear levels of DHX9, alters the intranuclear distribution of DHX9 (Figure 4). We have now performed additional experiments to further assess whether these proteins interact in the nucleoplasm.

1) As requested by the reviewers, we have fractionated nuclei derived from tissue culture cells into nucleoplasmic and nuclear envelope (NE) fractions and then immunopurified Nup98 and DHX9 from each fraction. We observed that Nup98 purified from nucleoplasm was associated with DHX9. By contrast, similar amounts of Nup98 immunopurified from NE fractions revealed only trace amounts of bound DHX9 (Figure 5). Consistent with these results, DHX9 purified from nucleoplasmic fractions was bound to Nup98.

2) We demonstrate that DHX9 binds a Nup98 mutant (GFP-Nup981-497) that is present in the nucleoplasmic but does not associate with NPCs (Figure 5—figure supplement 3).

3) DamID analysis was performed to examine whether Nup98 and DHX9 bind to similar regions of the genome, an observation that would support the conclusion that these proteins interact within the nucleoplasm. As shown in the revised manuscript (Figure 10 and Figure 10—figure supplement 1), DamID analysis detected Nup98 and DHX9 associated with overlapping loci. Moreover, the binding of each protein to these loci was dependent on the other (i.e. the chromatin interactions of Nup98 were reduced by depletion of DHX9, and vice versa). Together these multiple observations support the conclusion that Nup98 and DHX9 interact in the nucleoplasm.

Reviewer #2:

[…]

3) Figure 9 shows that interactions between specific mRNAs and Nup98 increase upon DHX9 depletion. Again, there seems to be a possibility that this is a consequence of altered mRNA export and accumulation of these mRNAs at NPC-bound Nup98. Furthermore, the decreased interaction between DHX9 and mRNAs upon depletion of Nup98 in 9B may be a consequence of decreased transcription of these mRNAs due to the function of Nup98 in gene activation. To help distinguish these possibilities, it is essential that the authors show total levels of these transcripts and preferably nascent transcripts in these conditions, in addition to the immunoprecipitated mRNA amounts shown in the figure. Furthermore, it would be helpful to determine if export of these mRNAs is disturbed upon Nup98 or DHX9 depletion, by single RNA FISH or by RT-PCR on nuclear vs. cytoplasmic fractions.

We have performed subcellular fractionation and examined levels of various mRNAs in the nuclear and cytoplasmic fractions from cells depleted of Nup98 or DHX9. We observed no significant alterations between the mock-depleted cells and cells depleted of Nup98 or DHX9 (Figure 9—figure supplement 2).

We have now included a figure showing levels of various transcripts examined in this manuscript (including E1A) following depletion of Nup98 or DHX9 (Figure 11). As discussed, we do observe changes in the levels of transcripts shown to bind Nup98 and DHX9, and we propose that these changes occur as a consequence of the role of these proteins in transcription. It is important to note that the change in the level of each specific mRNA bound to DHX9 or Nup98 is represented as a fold-change, between depleted and mock-depleted cells, in the ratio of bound:total mRNA of each species examined. Thus the fold changes in the ratios of bound:total amount of any given species account for changes in the levels of their transcript in the depleted cells.

You will note that depletion of Nup98 causes an increase in the binding of tested transcripts to DHX9, whereas depletion of DHX9 generally leads to a decrease in their binding to Nup98 (Figure 9). This is the reverse of what was reported in the initial submission due to an error in the calculation used, which has been thoroughly re- examined and corrected in the revised manuscript.

[Editors' note: further revisions were requested prior to acceptance, as described below.]

The manuscript has been improved and all three reviewers support publication. However, there are three very minor remaining issues that need to be addressed before acceptance, as outlined below:

1) Subsection “Nup98 stimulates the ATPase activity of DHX9”, the authors state that the basal ATPase rate of their DHX9 was comparable to previous publications. An attempt by reviewer 3 to calculate the rate in one of these citations (Zhang & Grosse), returned a value of ~15-30 s^-1^ (depending on the RNA used), quite different from the 1 s^-1^ in the current work. However, the other citation (Schütz et al) reported a rate constant very similar to the current work. The authors may want check this earlier work.

In two other references from the same research group (Zhang and Grosse, 1994; Zhang and Grosse, 1997), the basal ATPase rate they reported for DHX9 in the presence of poly IC RNA is 1.16 s^-1^ and 2.7 s^-1^. This is in good agreement with our results (1 s^-1^). We have changed the references.

2) The citation to "Aratani, 2006" should be the 2001 paper.

We have changed the references.

3) Subsection “Nup98 and DHX9 interact with a shared subset of mRNAs and gene loci.”, "Depletion of Nup98 or DHX9 significantly reduced the interactions of its binding partner with the target gene." This result here, with the DamID assay, seems not to fit in neatly with the result that "depletion of Nup98 led to a significant increase in the amount of each of the six mRNAs bound to DHX9." This isn't really dealt with directly by the authors. Is their model that the proteins bind cooperatively to DNA but bind RNA in a more complicated way, as noted "…consistent with a model in which DHX9 promotes the association of Nup98 with specific mRNAs, and Nup98 facilitates the release of these mRNAs from DHX9."

The short answer to this question is yes; we envisage a more complex model in which two distinct mechanisms of interaction are used by the Nup98/DHX9 complex to bind chromatin and RNA. We propose that Nup98 and DHX9 bind directly to RNA. However, we are not suggesting that Nup98 or DHX9 use a similar mechanism to bind DNA, but rather, based on previous reports, that they interact with chromatin through their association with CBP/p300 or other transcriptional regulators (subsection “Nup98-DHX9 complex functions to regulate transcription and mRNA processing”). These ideas are part of a more complex working model that is consistent with our various observations. However, we have not discussed the details of this broader model in the manuscript, as portions of it remain speculative. In this model, DHX9 is primarily responsible for locus-specific targeting of the Nup98/DHX9 complex. At these loci, the Nup98/DHX9 complex binds RNA transcripts. By stimulating DHX9 activity, Nup98 stimulates release of DHX9 from transcripts, preventing it from leaving the gene locus with the RNA transcript and, thus, effectively stabilizing the interactions of DHX9 with the chromatin. Our data showing that depletion of Nup98 increases the binding of DHX9 to specific RNAs and decreases DHX9 binding to their genes are consistent with this model. A discussion of aspects of this model, necessary to clarify the confusion experienced by the reviewer, is now included.